# MASKED GENERATIVE PRIORS IMPROVE WORLD MODELS SEQUENCE MODELLING CAPABILITIES

## ABSTRACT

Deep Reinforcement Learning (RL) has become the leading approach for creating artificial agents in complex environments. Model-based approaches, which are RL methods with world models that predict environment dynamics, are among the most promising directions for improving data efficiency, forming a critical step toward bridging the gap between research and real-world deployment. In particular, world models enhance sample efficiency by learning in imagination, which involves training a generative sequence model of the environment in a self-supervised manner. Recently, Masked Generative Modelling has emerged as a more efficient and superior inductive bias for modelling and generating token sequences. Building on the Efficient Stochastic Transformer-based World Models (STORM) architecture, we replace the traditional MLP prior with a Masked Generative Prior (e.g., MaskGIT Prior) and introduce GIT-STORM. We evaluate our model on two downstream tasks: reinforcement learning and video prediction. GIT-STORM demonstrates substantial performance gains in RL tasks on the Atari 100k benchmark. Moreover, we apply Categorical Transformer-based World Models to continuous action environments for the first time, addressing a significant gap in prior research. To achieve this, we employ a state mixer function that integrates latent state representations with actions, enabling our model to handle continuous control tasks. We validate this approach through qualitative and quantitative analyses on the DeepMind Control Suite, showcasing the effectiveness of Transformer-based World Models in this new domain. Our results highlight the versatility and efficacy of the MaskGIT dynamics prior, paving the way for more accurate world models and effective RL policies.

## 1 INTRODUCTION

Deep Reinforcement Learning (RL) has emerged as the premier method for developing agents capable of navigating complex environments. Deep RL algorithms have demonstrated remarkable performance across a diverse range of games, including arcade games (Mnih et al., 2015; Schrittwieser et al., 2020; Hafner et al., 2021; 2023), real-time strategy games (Vinyals et al., 2019; OpenAI, 2018), board games (Silver et al., 2016; 2018; Schrittwieser et al., 2020), and games with imperfect information (Schmid et al., 2021). Despite these successes, data efficiency remains a significant challenge, impeding the transition of deep RL agents from research to practical applications. Accelerating agent-environment interactions can mitigate this issue to some extent, but it is often impractical for real-world scenarios. Therefore, enhancing sample efficiency is essential to bridge this gap and enable the deployment of RL agents in real-world applications (Micheli et al., 2022).

Model-based approaches (Sutton & Barto, 2018) represent one of the most promising avenues for enhancing data efficiency in reinforcement learning. Specifically, models which learn a "world model" (Ha & Schmidhuber, 2018) have been shown to be effective in improving sample efficiency. This involves training a generative model of the environment in a self-supervised manner. These models can generate new trajectories by continuously predicting the next state and reward, enabling the RL algorithm to be trained indefinitely without the need for additional real-world interactions.

However, the effectiveness of RL policies trained in imagination hinges entirely on the accuracy of the learned world model. Therefore, developing architectures capable of handling visually complex and partially observable environments with minimal samples is crucial. Following Ha & Schmidhuber

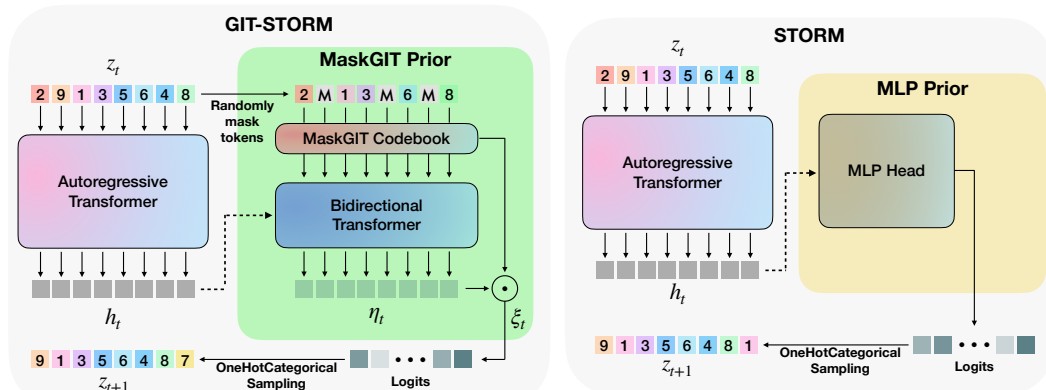

Figure 1: Overview of our proposed GIT-STORM method. (Left) The MaskGIT prior introduced to model the dynamics of the environment. The bidirectional transformer (Devlin et al., 2018) combines the hidden state given by the autoregressive transformer and the masked posterior $z_t \circ m_t$ to produce the prior corresponding to the next timestep. (Right) MLP prior originally used in STORM.

(2018), previous methods have employed recurrent neural networks (RNN) to model the dynamics of the environment Hafner et al. (2020; 2021; 2023). However, as RNNs impede parallelized computing due to their recurrent nature, some studies (Micheli et al., 2022; Robine et al., 2023; Zhang et al., 2023) have incorporated autoregressive transformer architectures (Vaswani et al., 2017) which have been shown to be effective across various domains, such as language (Devlin et al., 2019; Radford et al., 2019; Brown et al., 2020; Raffel et al., 2020), images (Dosovitskiy et al., 2021; He et al., 2022; Liu et al., 2023), and offline RL (Janner et al., 2021; Chen et al., 2021). For instance, IRIS (Micheli et al., 2022) utilize discrete autoencoders (Oord et al., 2017) to map raw pixels into a smaller set of image tokens to be used as the input to the world model, achieving super-human performance in ten different environments of the Atari 100k benchmark (Kaiser et al., 2019).

However, autoregressive transformers often suffer from hallucinations (Ji et al., 2023), where predicted states of the environment are unfeasible, deteriorating the agent's learning process. Additionally, their unidirectional generation process limits the ability to fully capture global contexts (Lee et al., 2022). To address these issues, TECO (Yan et al., 2023) introduces MaskGIT (Chang et al., 2022) prior $p_\phi(z_{t+1} \mid h_t)$, using a draft-and-revise algorithm to predict the next discrete representations in the sequence in video generation task. In-

Table 1: Comparison between an MLP prior and a spatial MaskGIT prior for video dynamics using Fréchet Video Distance (FVD).

|  | FVD ($\downarrow$) | |
| Method | DMLab | SSv2 |
| --- | --- | --- |
| TECO w/ MLP prior | 153 | 228 |
| TECO w/ MaskGIT prior | **48** | **199** |

terestingly, STORM shows that the latent representations $z_t$ have the biggest impact on the sequence modelling capabilities of the world model. Moreover, to the best of our knowledge, transformer-based world models have not yet been applied to continuous action environments (e.g., DeepMind Control Suite (DMC) (Kaiser et al., 2019)). The primary challenge lies in the reliance on categorical latent states, which are often ill-suited for representing continuous actions. Addressing this gap is critical for extending the applicability of transformer-based world models to a broader range of tasks.

In this paper, we introduce GIT-STORM, a novel world model inspired by STORM (Zhang et al., 2023), which leverages the MaskGIT prior to enhance world model sequence modelling capabilities. Building on insights from Yan et al. (2023), we demonstrate the superior performance of the MaskGIT prior over an MLP prior in predicting video dynamics, as evidenced by results in the DMLab (Beattie et al., 2016) and SSv2 (Goyal et al., 2017) datasets (Table 1). Here we summarize the main contributions of this work:

**C1:** We propose GIT-STORM, a novel world model that enhances STORM (Zhang et al., 2023) with a MaskGIT prior network for improved sequence modelling. Our model achieves state-of-the-art results on the Atari 100k benchmark, outperforming methods like DreamerV3 (Hafner et al., 2023) and IRIS, with comprehensive ablation studies showing the impact of discrete representation quality on downstream RL tasks.

Table 2: Comparison between the proposed GIT-STORM and relevant world models. AC stands for Actor Critic, OneHot for OneHotCategorical.

| Module | DreamerV3 (Hafner et al., 2023) | IRIS (Micheli et al., 2022) | TWM (Robine et al., 2023) | STORM (Zhang et al., 2023) | GIT-STORM (ours) |
|---|---|---|---|---|---|
| Latent space | [OneHot, Hidden] | VQ Codes | OneHot | OneHot | OneHot |
| Dynamics Model | RSSM | Transformer | TransformerXL | Transformer | Transformer |
| Dynamics Prior | MLP | MLP | MLP | MLP | MaskGIT |
| AC Input Space | [Latent, Hidden] | RGB | Latent | [Latent, Hidden] | [Latent, Hidden] |
| Experience Sampling | Uniform | Uniform | Balanced | Uniform | Uniform |

**C2:** We bridge the gap between transformer-based world models and continuous control tasks by using a State Mixer function that effectively combines categorical latent representations with continuous actions, enabling effective learning in continuous action spaces. Through rigorous evaluation on the DMC benchmark, we provide an in-depth analysis of the strengths and limitations of the proposed GIT-STORM model.

This paper marks a key step forward in extending transformer-based world models to more complex and diverse environments.

## 2 RELATED WORKS

### 2.1 MODEL-BASED RL: WORLD MODELS

Model-based RL has been a popular paradigm of reinforcement learning. With the advent of neural networks, it has become possible to model high-dimensional state spaces and thus, use model-based RL for environments with high-dimensional observations such as RGB images. In the last few years, based on PlaNet (Hafner et al., 2018), Hafner et al. proposed the Dreamer series (Hafner et al., 2020; 2021; 2023), a class of algorithms that learn the latent dynamics of the environment using a recurrent state space model (RSSM), while learning behavioral policy in the latent space. Currently, DreamerV3 (Hafner et al., 2023) has been shown to work across multiple tasks with a single configuration, setting the state-of-the-art across different benchmarks. The actor and critic in DreamerV3 learn from abstract trajectories of representations predicted by the world model.

With the advent of transformers (Vaswani et al., 2017) in sequence modelling and the promise of scaling performance across multiple tasks with more data, replacing the traditional RSSM backbones with transformer-based backbones has become a very active research direction. Although IRIS (Micheli et al., 2022), one of the first transformer-based world model approaches, obtains impressive results, its actor-critic operates in the RGB pixel space, making it almost 14x slower than DreamerV3. In contrast, methods such as TWM (Robine et al., 2023) and STORM (Zhang et al., 2023), use latent actor-critic input space. The proposed GIT-STORM employs it as well, as we believe it is the most promising direction to overcome sample efficiency constraints. More recently, STORM updated DreamerV3 by utilizing the transformer backbone. All aforementioned transformer-based world models use an MLP head to model a dynamics prior which is used to predict the discrete representation of the following timestep. In contrast, introduced by TECO (Yan et al., 2023), we employ a MaskGIT (Chang et al., 2022) prior head, which enhances the sequence modelling capabilities of the world model. Table 2 compares various design aspects of different world models. Furthermore, besides STORM, all the mentioned transformer-based world models concatenate the discrete action to the extracted categorical latent representations. As a result, none of these methods is able to handle continuous actions. In contrast, combining latent representations and actions with a state mixer, we successfully train STORM and GIT-STORM on a challenging continuous action environment (i.e., DMC).

### 2.2 MASKED MODELLING FOR VISUAL REPRESENTATIONS AND GENERATION

Inspired by the Cloze task (Taylor, 1953), BERT (Devlin et al., 2018) proposed a masked language model (MLM) pre-training objective that led to several state-of-the-art results on a wide class of natural language tasks. Following the success of BERT, Masked Autoencoders (MAEs) (He et al., 2022) learn to reconstruct images with masked patches during the pre-training stage. The learned representations are then used for downstream tasks. Zhang et al. (2021) similarly, improves upon a BERT-like masking objective for its non-autoregressive generation algorithm.

The most relevant to our work is MaskGIT (Chang et al., 2022), a non-autoregressive decoding approach that consists of a bidirectional transformer model, trained by learning to predict randomly masked visual tokens. By leveraging a bidirectional transformer (Devlin et al., 2018), it can better capture the global context across tokens during the sampling process. Furthermore, training on masked token prediction enables efficient, high-quality sampling at a significantly lower cost than autoregressive models. MaskGIT achieves state-of-the art performance on ImageNet dataset and achieves a $64\times$ speed-up on autoregressive decoding. The MaskGIT architecture has been applied to various tasks, such as video generation (Yan et al., 2023; Yu et al., 2023a;b) and multimodal generation (Mizrahi et al., 2024). For example, Yan et al. (2023) proposes TECO, a latent dynamics video prediction model that uses MaskGIT to model the prior for predicting the next timestep discrete representations, enhancing the sequence modelling of a backbone autoregressive transformer. Inspired by TECO, we adopt the use of MaskGIT prior for the world model, enhancing the sequence modelling capabilities, crucial for enabling and improving the agent policy learning behavior.

Further discussion of related works can be found in Appendix B.

## 3 METHOD

Following DreamerV3 (Hafner et al., 2023) and STORM (Zhang et al., 2023), we define our framework as a partially observable Markov decision process (POMDP) with discrete timesteps, $t \in \mathbb{N}$, scalar rewards, $r_t \in \mathbb{R}$, image observations, $o_t \in \mathbb{R}^{h \times w \times c}$, and discrete actions. $a_t \in \{1, \ldots, m_a\}$. These actions are governed by a policy, $a_t \sim \pi(a_t \mid o_{1:t}, a_{1:t-1})$, where $o_{1:t}$ and $a_{1:t-1}$ represent the previous observations and actions up to timesteps $t$ and $t - 1$, respectively. The termination of each episode is represented by a Boolean variable, $c_t \in \{0, 1\}$. The goal is to learn an optimal policy, $\pi$, that maximizes the expected total discounted rewards, $\mathbb{E}_\pi \left[ \sum_{t=1}^{\infty} \gamma^{t-1} r_t \right]$, where $\gamma \in [0, 1]$ serves as the discount factor. The learning process involves two parallel iterative phases: learning the observation and dynamics modules (World Model) and optimizing the policy (Agent).

In this section, we first provide an overview of the dynamics module of GIT-STORM. Then, we describe our dynamics prior head of the dynamics module, inspired by MaskGIT (Chang et al., 2022) (Figure 1). Finally, we explain the imagination phase using GIT-STORM, focusing on the differences between STORM and GIT-STORM. We follow STORM for the observation module and DreamerV3 for the policy definition, which are described in Appendix A.1 and A.2, respectively.

### 3.1 OVERVIEW: DYNAMICS MODULE

The dynamics module receives representations from the observation module and learns to predict future representations, rewards, and terminations to enable planning without the usage of the observation module (imagination). We implement the dynamics module as a Transformer State-Space Model (TSSM). Given latent representations from the observation module, $z_t$, and actions, $a_t$, the dynamics module predicts hidden states, $h_t$, rewards, $\hat{r}_t$, and episode termination flags, $\hat{c}_t \in \{0, 1\}$ as follows,

$$
\begin{align}
\zeta_t &= g_\theta(z_t, a_t) & \text{(State Mixer)} \\
h_t &= f_\theta(\zeta_{1:t}) & \text{(Autoregressive Transformer)} \\
z_{t+1} &\sim p_\phi(z_{t+1} \mid h_t) & \text{(Dynamics Prior Head)} \\
\hat{r}_t &\sim p_\phi(\hat{r}_t \mid h_t) & \text{(Reward Head)} \\
\hat{c}_t &\sim p_\phi(\hat{c}_t \mid h_t) & \text{(Termination Head)}
\end{align}
\tag{1}
$$

The world model is optimized to minimize the objective,

$$
\mathcal{L}(\phi) = \frac{1}{BT} \sum_{n=1}^{B} \sum_{t=1}^{T} \left[ \mathcal{L}_{\text{rew}}(\phi) + \mathcal{L}_{\text{term}}(\phi) + \beta_1 \mathcal{L}_{\text{dyn}}(\phi) + \beta_2 \mathcal{L}_{\text{rep}}(\phi) \right]
\tag{2}
$$

where $\beta_1, \beta_2$ are loss coefficients and $\mathcal{L}_{\text{rew}}(\phi), \mathcal{L}_{\text{term}}(\phi), \mathcal{L}_{\text{rep}}(\phi), \mathcal{L}_{\text{dyn}}(\phi)$ are reward, termination, representation, and dynamics losses, respectively. We use the symlog two-hot loss described in Hafner et al. (2023) as the reward loss. The termination loss is calculated as cross-entropy loss, $c_t \log \hat{c}_t + (1 - c_t) \log(1 - \hat{c}_t)$. In the following section, we define the dynamics prior in Eq. 1, as well as representation loss, $\mathcal{L}_{\text{rep}}$, and dynamics loss, $\mathcal{L}_{\text{dyn}}$.

## 3.2 Dynamics Prior Head: MaskGIT Prior

Given the expressive power of MaskGIT (Chang et al., 2022), we propose enhancing the dynamics module in the world model by replacing the current MLP prior with a MaskGIT prior, as shown in Figure 1. Given the posterior, $z_t$, and a randomly generated mask, $m \in \{0, 1\}^N$ with $M = \lceil \gamma N \rceil$ masked values where $\gamma = \cos\left(\frac{\pi}{2} t\right)$, the MaskGIT prior $p_\phi(z_{t+1} \mid h_t)$ is defined as follows.

First, the hidden states, $h_t$, are concatenated with the masked latent representations, $z_t \circ m_t$, where $\circ$ indicates element-wise multiplication. Despite $h_t$ being indexed by $t$, it represents the output of the $f_\theta$ and thus encapsulates information about the subsequent timestep. Consequently, the concatenation of $z_t$ and $h_t$ integrates information from both the current and the next timestep, respectively. A bidirectional transformer is then used to learn the relationships between these two consecutive representations, producing a summary representation, $\xi_t$. Finally, logits are computed as the dot product (denoted as $\odot$ in Figure 1) between the MaskGIT embeddings, which represent the masked tokens, and $\xi_t$. This dot product is also known as weight tying strategy, first formalized in Inan et al. (2017) and then used in the original MaskGIT Chang et al. (2022) and GPT-2 Radford et al. (2019) models as well because of its regularization effects that help preventing overfitting Inan et al. (2017). Indeed, this weight tying strategy (i.e., dot product) can be interpreted as a similarity distance between the embeddings and $\xi_t$. Indeed, from a geometric perspective, both cosine similarity and the dot product serve as similarity metrics, with cosine similarity focusing on the angle between two vectors, while the dot product accounts for both the angle and the magnitude of the vectors. Therefore, by optimizing the MaskGIT prior, this dot product aligns the embeddings with $\xi_t$, thereby facilitating and improving the computation of logits. In contrast, when using the MLP prior, the logits are generated as the output of an MLP that only takes $h_t$ as input. This approach requires the model to learn the logits space and their underlying meaning without any inductive bias, making the learning process more challenging.

During training, we follow the KL divergence loss of DreamerV3 (Hafner et al., 2023), which consists of two KL divergence losses which differ in the stop-gradient operator, $\mathrm{sg}(\cdot)$, and loss scale. We account for the mask tokens in the posterior and define $\mathcal{L}_{\mathrm{dyn}}$ and $\mathcal{L}_{\mathrm{rep}}$ as,

$$\mathcal{L}_{\mathrm{dyn}}(\phi) \doteq \max\big(1, \mathrm{KL}\big[\, \mathrm{sg}(q_\phi(z_t \mid x_t)) \circ m_t \,\|\, p_\phi(z_t \mid h_{t-1}) \,\big]\big) \tag{3}$$

$$\mathcal{L}_{\mathrm{rep}}(\phi) \doteq \max\big(1, \mathrm{KL}\big[\, q_\phi(z_t \mid x_t) \circ m_t \,\|\, \mathrm{sg}(p_\phi(z_t \mid h_{t-1}))\big]\big) \tag{4}$$

where $m_t$ is multiplied element-wise with the posterior, eliminating the masked tokens from the loss.

**Sampling.** During inference, since MaskGIT has been trained to model both unconditional and conditional probabilities, we can sample any subset of tokens per sampling iteration. Following Yan et al. (2023), we adopt the Draft-and-Revise decoding scheme introduced by Lee et al. (2022) to predict the next latent state (Algorithm 1 and 2). During the draft phase, we initialize a partition $\Pi$ which contains $T_{\mathrm{draft}}$ disjointed mask vectors $\mathbf{m}$ of size (latent dim $\div\ T_{\mathrm{draft}}$), which together mask the whole latent representation. Iterating through all mask vectors in $\Pi$, the resulting masked representations are concatenated with the hidden states $h_t$ from Eq. 1 and fed to the MaskGIT prior head that computes the logits of the tokens correspondent to $h_t$ and $\mathbf{m}^i$. Such logits are then used to sample the new tokens that replace the positions masked by $\mathbf{m}^i$. During the revise phase, the whole procedure is repeated $\Gamma$ times. As a result, when sampling the new tokens, the whole representation is taken into account, resulting in a more consistent and meaningful sampled state.

## 3.3 State Mixer for Continuous Action Environments

When using a TSSM as the dynamics module, the conventional approach has been to concatenate discrete actions with categorical latent representations and feed this sequence into the autoregressive transformer. However, this method is ineffective for continuous actions, as one-hot categorical representations or VQ-codes (Oord et al., 2017) are poorly suited for representing continuous values. To overcome this limitation, we repurpose the state mixer function $g_\theta(\cdot)$ introduced in STORM, which combines the latent representation and the action into a unified mixed representation $\zeta_t$. This approach allows for the integration of both continuous and discrete actions with latent representations, enabling the application of TSSMs to environments that require continuous action spaces.

**Algorithm 1** Draft-and-Revise decoding scheme
___
**Require:** Partition sampling distributions $p_{\text{draft}}$ and $p_{\text{revise}}$, the number of revision iterations $\Gamma$, hidden states $h_t$, model $\theta$
     /* draft phase */
 1: $\boldsymbol{z}^{\text{empty}} \leftarrow \big([\texttt{MASK}], \cdots, [\texttt{MASK}]\big)^N$
 2: $\boldsymbol{\Pi} \sim p_{\text{draft}}(\boldsymbol{\Pi}; T_{\text{draft}})$
     /* generate a draft prior map */
 3: $\boldsymbol{z}^0 \leftarrow \text{MASKGIT HEAD}(\boldsymbol{z}^{\text{empty}}, \boldsymbol{\Pi}, h_t; \theta)$
     /* revision phase */
 4: **for** $\gamma = 1, \cdots, \Gamma$ **do**
 5:     $\boldsymbol{\Pi} \sim p_{\text{revise}}(\boldsymbol{\Pi}; T_{\text{revise}})$
 6:     $\boldsymbol{z}^\gamma \leftarrow \text{MASKGIT HEAD}(\boldsymbol{z}^{\gamma-1}, \boldsymbol{\Pi}, h_t; \theta)$
 7: **end for**
 8: $\boldsymbol{z}_{t+1} \leftarrow \boldsymbol{z}^\Gamma$
 9: **return** $\boldsymbol{z}_{t+1}$
___

**Algorithm 2** MASKGIT HEAD
___
**Require:** Generated latents $\boldsymbol{z}$, hidden states $h_t$, partition $\boldsymbol{\Pi} = (\mathbf{m}^1, \ldots, \mathbf{m}^T)$, model $\theta$
 1:                                                ▷ Update the codes
 2: **for** $i = 1$ to $T$ **do**
 3:     MaskGIT_codes $\leftarrow$ MaskGIT_Codebook$(\boldsymbol{z} \circ \mathbf{m}^i)$
 4:     $\xi \leftarrow$ BidirectionalTransformer(MaskGIT_codes, $h_t$)
 5:     logits $\leftarrow \xi \odot$ MaskGIT_embeddings
 6:     $\hat{\boldsymbol{z}} \sim$ Categorical(logits)
 7:     $\boldsymbol{z} \leftarrow (1 - \mathbf{m}^i) \circ \boldsymbol{z} + \mathbf{m}^i \circ \hat{\boldsymbol{z}}$
 8: **end for**
 9: **return** $\boldsymbol{z}$
___

### 3.4 IMAGINATION PHASE

Instead of training the policy by interacting with the environment, model-based approaches use the learned representation of the environment and plan in imagination (Hafner et al., 2018). This approach allows sample-efficient training of the policy by propagating value gradients through the latent dynamics. The interdependence between the dynamics generated by the world model and agent's policy makes the quality of the imagination phase crucial for learning a meaningful policy. The imagination phase is composed of two phases, conditioning phase and the imagination one. During the conditioning phase, the discrete representations $z_t$ are encoded and fed to the autoregressive transformer. The conditioning phase gives context for the imagination one, using the cached keys and values (Yan et al., 2021) computed during the conditioning steps.

Differently from STORM, which uses a MLP prior to compute the next timestep representations, we employ MaskGIT to accurately model the dynamics of the environment. By improving the quality of the predicted trajectories, the agent is able to learn a superior policy.

## 4 EXPERIMENTS

In this section, we analyse the performance of GIT-STORM and its potential limitations by exploring the following questions: (a) How does the MaskGIT Prior affect TSSMs learning behavior and performances on related downstream tasks (e.g., Model-based RL and Video Prediction tasks)? (b) Can Transformer-based world models learn to solve tasks on continuous action environments when using state mixer functions?

### 4.1 EXPERIMENTAL SETUP

To evaluate and analyse the proposed method, we consider both discrete and continuous actions environments, namely Atari 100k benchmark (Kaiser et al., 2019) and DeepMind Control Suite (Tassa et al., 2018) respectively. On both environments, we conduct both RL and video prediction tasks.

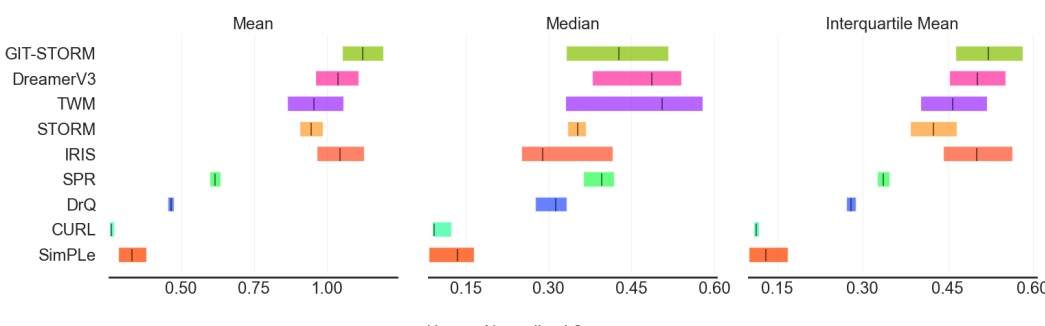

Figure 2: (Left) Human normalized mean, across the Atari 100k benchmark. GIT-STORM outperforms all other baselines. (Middle) Human normalized median. TWM achieves the highest median value of $51\%$. (Right) IQM. GIT-STORM outperforms all other baselines.

**Benchmark and baselines.** Atari 100k benchmark consists of 26 different video games with discrete action space. The constraint of 100k interactions corresponds to a total of 400k frames used for training, as frame skipping is set to 4. For RL task on Atari 100k benchmark, we compare GIT-STORM against one model-free method, SimPLe (Kaiser et al., 2019), one RSSM, DreamerV3 (Hafner et al., 2023), and three TSSM models (i.e., IRIS (Micheli et al., 2022), TWM (Robine et al., 2023), and STORM (Zhang et al., 2023)). DMC benchmark consists of 18 control tasks with continuous action space. We restrict the models to be trained with only 500k interactions (1M frames) by setting frame skipping to 2. For RL task on DMC benchmark, we compare our model against SAC (Haarnoja et al., 2018), CURL (Laskin et al., 2020), DrQ-v2 (Yarats et al., 2022), PPO (Schulman et al., 2017), DreamerV3 (Hafner et al., 2023), and STORM (Zhang et al., 2023). We trained GIT-STORM on 5 different seeds. For video prediction tasks, we compare GIT-STORM with STORM only to understand how the MaskGIT Prior affects the visual quality of predicted frames and its influence on the policy training.

Extended details of the baselines for both benchmarks can be found in Appendix J.

**Evaluation metrics.** Proper evaluation of RL algorithms is known to be difficult due to both the stochasticity and computational requirements of the environments (Agarwal et al., 2021). To provide an accurate evaluation of the models, we consider a series of metrics to assess the performances of the considered baselines on across the selected experiments. We report human normalized mean and median as evaluation metrics, aligning with prior literature. We also report interquartile Mean (IQM), Optimality Gap, Performance Profiles (scores distributions), and Probability of Improvement (PI), which provide a statistically grounded perspective on the model evaluation (Agarwal et al., 2021).

For video prediction task, we report two metrics: Fréchet Video Distance (FVD) (Unterthiner et al., 2019) to evaluate visual quality of the predicted frames, and perplexity (Jelinek et al., 2005) measure of the predicted tokens to evaluate the token utilization by the dynamics prior head. We use the trained agent to collect ground truth episodes and use the world model to predict the frames. We report the FVD over 256 videos which are conditioned on the first 8 frames to predict 48 frames.

A full description of these metrics can be found in Appendix K.

## 4.2 Results on Discrete Action Environments: Atari 100k

**RL task.** Figure 2 summarizes the human normalized mean and median, and IQM score. The full results on individual environments can be found in the Appendix due to space limitations (Table 5). We can see that while TWM and DreamerV3 present a higher human median than GIT-STORM (TWM: $51\%$, DreamerV3: $49\% \rightarrow$ GIT-STORM: $42.6\%$), GIT-STORM dominates in terms of human mean (TWM: $96\%$, DreamerV3: $104\% \rightarrow$ GIT-STORM: $112.6\%$). In terms of IQM, a more robust and statistically meaningful metric, GIT-STORM significantly outperforms the related baselines (DreamerV3: 0.501, IRIS: 0.502 $\rightarrow$ GIT-STORM: 0.522).

Figure 3 (Left) compares PI against the baselines. Noticeably, GIT-STORM presents $PI > 0.5$ for all baselines, which indicates that, from a probabilistic perspective GIT-STORM would outperform

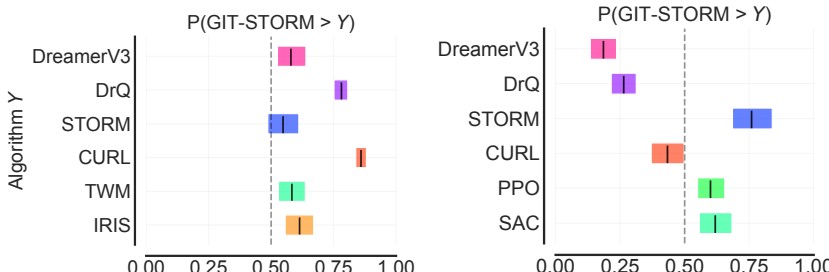

Figure 3: Probability of Improvement of the mentioned baselines and GIT-STORM in the Atari 100k benchmark (Left) and DMC benchmark (Right). The results represent how likely it is for GIT-STORM to outperform other baselines.

each baseline on a random task Y from Atari 100k with a probability greater than $0.5$. Figure 8 illustrates the Optimality Gap, while Figure 9 presents the fraction of runs with score $> \tau$ for different human normalized scores; both confirm the trends observed so far. Moreover, a closer look to Table 5 reveals that GIT-STORM presents an optimality gap of $0.500$, marginally beating DreamerV3, which reports $0.503$ and significantly outperforming all other baselines.

**Video Prediction task.** Table 3 shows video prediction results on selected Atari 100k environments. The table shows that GIT-STORM presents, on average, lower FVD and higher perplexity than STORM (e.g., in Freeway, STORM: 105.45, 33.15 → GIT-STORM: 80.33, 67.92, respectively). Figure 5 shows several video prediction results on each environment. For example, on Boxing, we can see that GIT-STORM is able to predict more accurately into the future. The differences in the other two games are smaller, as the player in each game has a much smaller dimension. We think GIT-STORM achieves higher perplexity because the learned agent can collect more diverse episodes.

## 4.3 RESULTS ON CONTINUOUS ACTION ENVIRONMENTS: DEEPMIND CONTROL SUITE

**RL task.** Figure 4 summarizes the human normalized mean and median, and IQM score. The full results on individual environments can be found in the Appendix (Table 6). Although DreamerV3 outperforms all other models on average, Table 6 shows that GIT-STORM presents state-of-the-art scores on two environments, Walker Stand and Quadruped Run. Compared to STORM, GIT-STORM consistently and significantly outperforms across the whole benchmark in terms of human median and mean (STORM: 31.50, 214.50 → GIT-STORM: 475.12, 442.10, respectively). For PI, GIT-STORM achieves $PI > 0.5$ than STORM, PPO, and SAC (e.g., GIT-STORM: 0.75, 0.60 and 0.63, over STORM, PPO and SAC, respectively) (Figure 3 (Right)).

**Video Prediction task.** Table 4 shows video prediction results on selected DMC environments. The table shows that our model achieves lower FVD and higher perplexity than STORM for all environments. The video prediction results in Figure 6 show that although both models fail to capture the dynamics accurately, GIT-STORM generates marginally better predictions, leading to higher perplexity as well.

## 5 DISCUSSION

The proposed GIT-STORM uses a Masked Generative Prior (MaskGIT) to enhance the world model sequence modelling capabilities. Indeed, as discussed in the introduction, high quality and accurate representations are essential to guarantee and enhance agent policy learning in imagination. Remark-

Table 3: FVD and perplexity comparisons of STORM and GIT-STORM on selected Atari 100k environments.

| Game | FVD ($\downarrow$) | | Perplexity ($\uparrow$) | |
|---|---|---|---|---|
| | STORM | GIT-STORM | STORM | GIT-STORM |
| Boxing | **1458.32** | 1580.32 | 49.24 | **54.95** |
| Hero | 381.16 | **354.16** | 10.55 | **30.25** |
| Freeway | 105.45 | **80.33** | 33.15 | **67.92** |

Table 4: FVD and perplexity comparisons of STORM and GIT-STORM on selected DMC environments.

| Task | FVD ($\downarrow$) | | Perplexity ($\uparrow$) | |
|---|---|---|---|---|
| | STORM | GIT-STORM | STORM | GIT-STORM |
| Cartpole Balance Sparse | 2924.81 | **1892.44** | 1.00 | **3.76** |
| Hopper Hop | 4024.11 | **3458.19** | 3.39 | **22.59** |
| Quadruped Run | 3560.33 | **1000.91** | 1.00 | **2.61** |

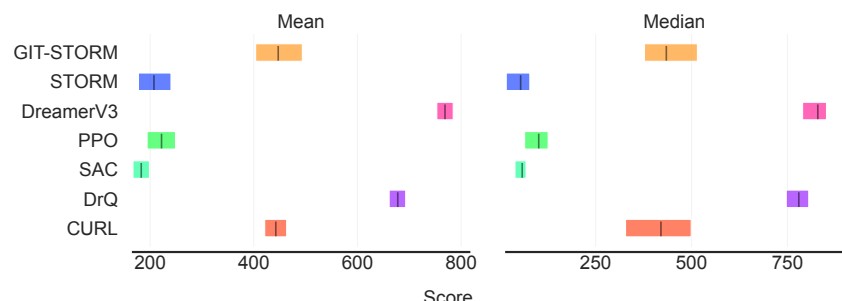

Figure 4: Comparison of human normalized mean (left) and median (right) on DMC benchmark.

ably, the proposed GIT-STORM is the only world model, among the ones that use uniform sampling and latent actor critic input space, that is able to achieve non-zero reward on the Freeway environment (e.g., DreamerV3: 0, STORM: $0 \rightarrow$ GIT-STORM: 13). Indeed, both STORM and IRIS resorted in ad-hoc solutions to get positive rewards, such as changing the sampling temperature (Micheli et al., 2022) and using demonstration trajectories (Zhang et al., 2023). Such result, together with the quantitative results on the Atari 100k and DMC benchmarks, clearly answer question (a) - the presented MaskGIT prior improves the policy learning behavior and performance on downstream tasks (e.g., Model-based RL and Video Prediction) of TSSMs. Moreover, the FVD and perplexity comparisons in Table 3 and Table 4 suggest that GIT-STORM has better predictive capabilities, learns a better dynamics module, and presents more accurate imagined trajectories (Figure 5, Figure 6). Similarly to image synthesis (Chang et al., 2022) and video prediction (Yu et al., 2023a) tasks, we show how using masked generative modelling is a better inductive bias to model the prior dynamics of discrete representations and improve the downstream usefulness of world models on RL tasks. Furthermore, the MaskGIT Prior can be used in any sequence modelling backbone that uses categorical latent representations (e.g., VideoGPT (Yan et al., 2021), IRIS (Micheli et al., 2022)), positioning itself as a very versatile approach. In this work we do not apply a MaskGIT prior on top of IRIS only because of computational and time constraints - IRIS requires 168h of training on a V100 GPU for a single run (Zhang et al., 2023) .

Noticeably, the quantitative results on DMC benchmarks answer question (b) - It is possible to train TSSMs when using a mixer function to combine categorical representations and continuous actions. Indeed, both STORM and GIT-STORM are able to learn meaningful policies within the DMC benchmark. Remarkably, GIT-STORM outperforms STORM with an substantial margin, while

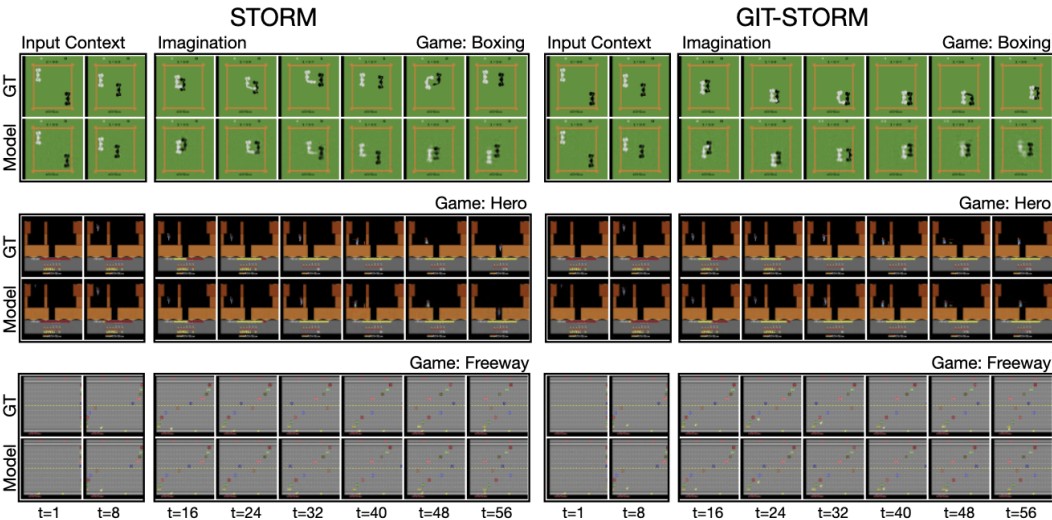

Figure 5: Generated trajectories of STORM and GIT-STORM on selected Atari 100k environments. The model uses the first 8 frames as context and then generates the following 48 frames.

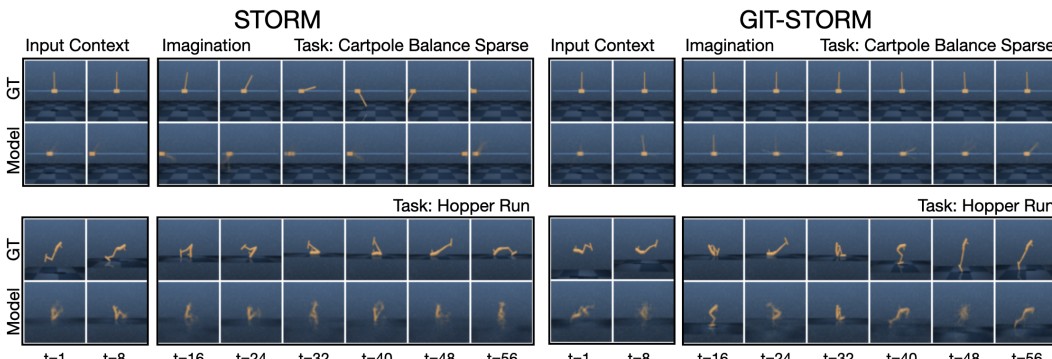

Figure 6: Generated trajectories of STORM and GIT-STORM on selected DMC environments. The model uses the first 8 frames as context and then generates the following 48 frames.

using exactly the same policy learning algorithm. Interestingly, Figure 15 presents an ablation of the used state mixer function, revealing that the overall learning behavior highly depends on the used inductive bias. Surprisingly, the simplest one (e.g., concatenation of $z_t$ and $a_t$) is the only one that works meaningfully. We leave the exploration of better inductive biases (e.g., imposing specific information bottlenecks (Meo et al., 2024a)) to improve the state mixer function as future work.

**Limitations and Future Work.** The current implementation has been validated on environments that do not require extensive training steps (e.g., ProcGen (Cobbe et al., 2020), Minecraft (Kanitscheider et al., 2021)) to be trained. We keep as a future work the validation of GIT-STORM on ProcGen and Minecraft environments. As suggested by Yan et al. (2023), using a MaskGIT prior could benefit the world model learning behavior in a visually challenging environment like Minecraft. From a technical point of view, one of the main limitations of the proposed world model is that we use only one iteration for the Draft-and-Revise decoding scheme (Lee et al., 2022). Indeed, while using one iteration speeds up training and evaluation, we do not fully exploit the advantages of this decoding scheme. As a result, in environments like Pong or Breakout, which present small objects (e.g., white or red balls, respectively), using a masked generative approach can lead to filtering such objects out, degrading the downstream performances in these environments. The main reason is that the presented decoding scheme scales exponentially with the number of iterations. We leave as future work the definition of a decoding scheme that scales more efficiently with the number of iterations.

## 6 CONCLUSION

The motivation for this work stems from the need to improve the quality and accuracy of world models representations in order to enhance agent policy learning in challenging environments. Inspired by (Yan et al., 2023), we conducted experiments using the TECO framework on video prediction tasks with DMLab and SSv2 (Goyal et al., 2017) datasets. Replacing an MLP prior with a MaskGIT (Chang et al., 2022) prior significantly improved the sequence modelling capabilities and the related performance on the video prediction downstream task. Building upon these insights, we proposed GIT-STORM, which employs a MaskGIT Prior to enhance the sequence modelling capabilities of world models, crucial to improve the policy learning behavior (Micheli et al., 2022). Moreover, through the use of a state mixer function, we successfully combined categorical latent representations with continuous actions, and learned meaningful policies on the related environments. We validated the proposed approach on the Atari 100k and the DMC benchmarks. Our quantitative analysis showed that GIT-STORM on average outperforms all baselines in the Atari 100k benchmark while outperforming STORM with a significant margin on the DMC benchmark. Although our approach does not beat the state-of-the-art in the DMC benchmark, the presented quantitative and qualitative evaluations led to the conclusion that masked generative priors (e.g., MaskGIT Prior) improve world models sequence modelling capabilities and the related downstream usefulness.

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

# A    GIT-STORM FRAMEWORK

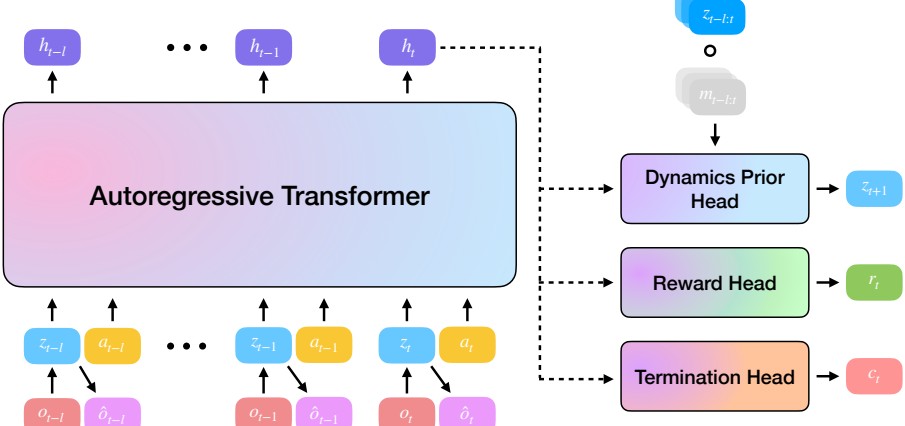

Figure 7: GIT-STORM End-to-End pipeline. Similar to STORM (Zhang et al., 2023), GIT-STORM performs sequence modelling using an autoregressive transformer, which predicts future stochastic latents, $z_t$, reward, $r_t$ and termination, $c_t$. In contrast with STORM, GIT-STORM uses a Masked Generative Prior to model the dynamics of the environment.

Similar to previous TSSM-based world models (Micheli et al., 2022; Chen et al., 2022; Zhang et al., 2023), GIT-STORM consists of a world model with two modules, VAE-based observation module and autoregressive dynamics module, and a policy trained in the latent space. Figure 7 describes the world model architecture of GIT-STORM. In the following sections, we provide details of the observation module and policy.

## A.1    OBSERVATION MODULE

Following STORM, the observation module is a variational autoencoder (VAE) (Kingma & Welling, 2013), which encodes observations, $o_t$, into stochastic latent representations, $z_t$, and decodes back the latents to the image space, $\hat{o}_t$:

$$\text{Observation encoder: } z_t \sim q_\phi(z_t \mid o_t) \tag{5}$$
$$\text{Observation decoder: } \hat{o}_t = p_\phi(z_t) \tag{6}$$

The observations are encoded using a convolutional neural network (CNN) encoder (LeCun et al., 1989) which outputs the logits used to sample from a categorical distribution. The distribution head applies an unimix function over the computed logits to prevent the probability of selecting any category from being zero (Sullivan et al., 2023). Since the sampled latents lack gradients, we use the straight-through gradients trick (Bengio et al., 2013) to preserve them. The decoder, modeled using a CNN, reconstructs the observation from the latents, $z_t$. While the encoder is updated using gradients coming from both observation and dynamics modules, the decoder is optimized using only the Mean Squared Error (MSE) between input and reconstructed frames:

$$\mathcal{L}_{\text{Observation Model}} = \text{MSE}(o_t, \hat{o}_t) \tag{7}$$

## A.2    POLICY LEARNING

Following the model-based RL research landscape (DreamerV3; Hafner et al., 2023) we cast the agent policy learning framework using the actor-critic approach (Mnih et al., 2016). The agent actor-critic is trained purely from agent state trajectories $s_t = [z_t, h_t]$ generated by the world model. The actor aims to learn a policy that maximizes the predicted sum of rewards and the critic aims to predict the

distribution of discounted sum of rewards by the current actor:

$$\text{Actor: } a_t \sim \pi_\theta(a_t|s_t) \text{ , Critic: } V_\psi(s_t) \approx \mathbb{E}_{\pi_\theta, p_\phi} \left[ \sum_{\tau=0}^{\infty} \gamma^\tau r_{t+\tau} \right], \tag{8}$$

where $\gamma$ is a discount factor.

We follow the setup of STORM (Zhang et al., 2023) and DreamerV3 (Hafner et al., 2023) to train the agent. First, a random trajectory is sampled from the replay buffer to compute the initial state of the agent. Then, using the sampled trajectory as context, the world model and actor generate a trajectory of imagined model states, $s_{1:L}$, actions, $a_{1:L}$, rewards, $\hat{r}_{1:L}$, and termination flags, $\hat{c}_{1:L}$, where $L$ is the imagination horizon. To estimate returns that consider rewards beyond the prediction horizon, we compute bootstrapped $\lambda$-returns (Sutton & Barto, 1998; Hafner et al., 2023) defined recursively as follows:

$$G_l^\lambda = \hat{r}_l + \gamma \hat{c}_l \left[ (1-\lambda)V_\psi(s_{l+1}) + \lambda V_{l+1}^\lambda \right] \text{ , } G_L^\lambda = V_\psi(s_L) \tag{9}$$

To stabilize training and prevent the model from overfitting, we regularize the critic towards predicting the exponential moving average (EMA) of its own parameters. The EMA of the critic is updated as,

$$\psi_{l+1}^{\text{EMA}} = \sigma \psi_l^{\text{EMA}} + (1-\sigma)\psi_l, \tag{10}$$

where $\sigma$ is the decay rate. As a result, the critic learns to predict the distribution of the return estimates using the following maximum likelihood loss:

$$\mathcal{L}_\psi = \frac{1}{BL} \sum_{n=1}^{B} \sum_{l=1}^{L} \left[ (V_\psi(s_l) - \text{sg}(G_l^\lambda))^2 + (V_\psi(s_l) - \text{sg}(V_{\psi^{\text{EMA}}}(s_l)))^2 \right], \tag{11}$$

The actor learns to choose actions that maximize return while enhancing exploration using an entropy regularizer (Williams & Peng, 1991; Hafner et al., 2023). Reinforce estimator (Williams, 1992) is used for actions, resulting in the surrogate loss function:

$$\mathcal{L}_\theta = \frac{1}{BL} \sum_{n=1}^{B} \sum_{l=1}^{L} \left[ -\text{sg}\left( \frac{G_l^\lambda - V_\psi(s_l)}{\max(1,S)} \right) \ln \pi_\theta(a_l|s_l) - \eta H(\pi_\theta(a_l|s_l)) \right], \tag{12}$$

where $\text{sg}(\cdot), H(\cdot)$ are stop gradient operator and entropy, respectively, and $\eta$ is a hyperparameter coefficient of the entropy loss. When training the actor, the rewards are computed between the range from the $5^{\text{th}}$ to the $95^{\text{th}}$ percentile and smoothed out by using an EMA to be robust to outliers. Therefore, the normalization ratio $S$ is,

$$S = \text{EMA}(\text{percentile}(G_l^\lambda, 95) - \text{percentile}(G_l^\lambda, 5)). \tag{13}$$

## B  EXTENDED RELATED WORKS: VIDEO PREDICTION MODELLING

Video prediction, a fundamental task in computer vision, aims to generate or predict sequences of future frames based on conditioning past frames. The downstream tasks of video prediction modelling span a wide range of domains, showcasing its significance in different fields, such as autonomous driving (Hu et al., 2023), robot navigation (DeSouza & Kak, 2002) controllable animation (Mahapatra & Kulkarni, 2022), weather forecasting (Bi et al., 2023; Meo et al., 2024b), and model based reinforcement learning (Hafner et al., 2018; 2020; 2021; 2023; Zhang et al., 2023; Micheli et al., 2022). Video prediction modelling is known for its sample inefficiency, which poses significant challenges in learning accurate and reliable models in a feasible time (Ming et al., 2024). To address this, recent advancements have introduced spatio-temporal state space models, which typically consist of a feature extraction component coupled with a dynamics prediction module. These models aim to understand and predict the evolution of video frames by capturing both spatial and temporal relationships. Notable examples include NUWÄ (Wu et al., 2022) and VideoGPT (Yan et al., 2021) which respectively use 2D and 3D convolutional layers to extract the latent representations and an autoregressive transformer to perform sequence modelling in the latent space. Moreover, TECO (Yan et al., 2023) introduces the use of MaskGIT (Chang et al., 2022) prior to improve the accuracy of the predicted discrete latents and uses a 1D convolution to enhance temporal consistency. Furthermore, VideoPoet (Kondratyuk et al., 2023), which is able to handle multiple modalities and perform a variety of tasks besides video prediction.

# C   FULL RESULTS ON RL TASK

In this section we report and present the full evaluation and comparison on the two RL benchmark environments, Atari 100k (Kaiser et al., 2019) and DMC (Tassa et al., 2018). Table 5 and Table 6 are the results on Atari 100k and DMC, respectively.

Table 5: Evaluation on the 26 games in the Atari 100k benchmark. We report mean scores as well as aggregated human normalized mean and median, Interquantile Mean (IQM), and Optimality Gap. Following the conventions of Hafner et al. (2021), scores that are the highest or within $5\%$ of the highest score are highlighted in bold.

| Game | Rand | Hum | SimPLe reported | TWM reported | IRIS reported | DreamerV3 reproduced | STORM reproduced | GIT-STORM ours |
|---|---|---|---|---|---|---|---|---|
| Alien | 228 | 7128 | 617 | 675 | 420 | 804 | **1364** | 1145 |
| Amidar | 6 | 1720 | 74 | 122 | 143 | 122 | **239** | 181 |
| Assault | 222 | 742 | 527 | 683 | **1524** | 642 | 707 | 967 |
| Asterix | 210 | 8503 | 1128 | 1116 | 854 | **1190** | 865 | 811 |
| Bank Heist | 14 | 753 | 34 | 467 | 53 | **752** | 375 | 503 |
| Battle Zone | 2360 | 37188 | 4031 | 5068 | **13074** | 11600 | 10780 | 9470 |
| Boxing | 0 | 12 | 8 | **78** | 70 | 71 | **80** | **81** |
| Breakout | 2 | 30 | 16 | 20 | **84** | 24 | 12 | 12 |
| Chopper Command | 811 | 7388 | 979 | 1697 | 1565 | 680 | 2293 | 2048 |
| Crazy Climber | 10780 | 35829 | 62584 | 71820 | 59234 | **86000** | 54707 | 55237 |
| Demon Attack | 152 | 1971 | 208 | 350 | **2034** | 203 | 229 | 223 |
| Freeway | 0 | 30 | 17 | 24 | **31** | 0 | 0 | 13 |
| Frostbite | 65 | 4335 | 237 | **1476** | 259 | 1124 | 646 | 582 |
| Gopher | 258 | 2413 | 597 | 1675 | 2236 | 4358 | 2631 | **8562** |
| Hero | 1027 | 30826 | 2657 | 7254 | 7037 | 12070 | 11044 | **13351** |
| Jamesbond | 29 | 303 | 101 | 362 | 463 | 290 | **552** | 471 |
| Kangaroo | 52 | 3035 | 51 | 1240 | 838 | **4080** | 1716 | 1601 |
| Krull | 1598 | 2666 | 2204 | 6349 | 6616 | **7326** | 6869 | **7011** |
| Kung Fu Master | 256 | 22736 | 14862 | **24555** | 21760 | 19100 | 20144 | **24689** |
| Ms Pacman | 307 | 6952 | 1480 | 1588 | 999 | 1370 | **2673** | 1877 |
| Pong | -21 | 15 | 13 | **19** | 15 | **19** | 8 | 6 |
| Private Eye | 25 | 69571 | 35 | 87 | 100 | 140 | **2734** | 2225 |
| Qbert | 164 | 13455 | 1289 | 3331 | 746 | 1875 | 2986 | **3924** |
| Road Runner | 12 | 7845 | 5641 | 9109 | 9615 | 14613 | 12477 | **17449** |
| Seaquest | 68 | 42055 | 683 | **774** | 661 | 571 | 525 | 459 |
| Up N Down | 533 | 11693 | 3350 | **15982** | 3546 | 7274 | 7985 | 10098 |
| Human Mean (↑) | 0% | 100% | 33% | 96% | 105% | 104% | 94.7% | **112.6%** |
| Human Median (↑) | 0% | 100% | 13% | **51%** | 29% | 49% | 35.7% | 42.6% |
| IQM (↑) | 0.00 | 1.00 | 0.130 | 0.459 | 0.501 | 0.502 | 0.426 | **0.522** |
| Optimality Gap (↓) | 1.00 | 0.00 | 0.729 | 0.513 | 0.512 | 0.503 | 0.528 | **0.500** |

Table 6: Evaluation on the DeepMind Control Suite benchmark. We report scores under visual inputs at 1M frames as well as aggregated human normalized mean and median. Following the conventions of Hafner et al. (2021), scores that are the highest or within 5% of the highest score are highlighted in bold.

| Task | SAC | CURL | PPO | DrQ-v2 | DreamerV3 | STORM | GIT-STORM |
|---|---|---|---|---|---|---|---|
| Acrobot Swingup | 5.1 | 5.1 | 2.3 | 128.4 | **210.0** | 12.2 | 2.1 |
| Cartpole Balance | **963.1** | **979.0** | 507.3 | **991.5** | **996.4** | 208.9 | 567.0 |
| Cartpole Balance Sparse | **950.8** | **981.0** | 890.4 | **996.2** | **1000.0** | 15.2 | 790.9 |
| Cartpole Swingup | 692.1 | 762.7 | 259.9 | **858.9** | **819.1** | 124.8 | 452.2 |
| Cartpole Swingup Sparse | 154.6 | 236.2 | 0.0 | 706.9 | **792.9** | 0.6 | 97.3 |
| Cheetah Run | 27.2 | 474.3 | 95.5 | 691.0 | **728.7** | 137.7 | 552.5 |
| Cup Catch | 163.9 | **965.5** | 821.4 | **931.8** | **957.1** | 735.5 | 841.5 |
| Finger Spin | 312.2 | **877.1** | 121.4 | **846.7** | 818.5 | 753.8 | 787.0 |
| Finger Turn Easy | 176.7 | 338.0 | 311.0 | 448.4 | **787.7** | 307.3 | 334.1 |
| Finger Turn Hard | 70.5 | 215.6 | 0.0 | 220.0 | **810.8** | 1.4 | 148.6 |
| Hopper Hop | 3.1 | 152.5 | 0.3 | 189.9 | **369.6** | 0.0 | 193.6 |
| Hopper Stand | 5.2 | 786.8 | 6.6 | **893.0** | **900.6** | 0.0 | 664.6 |
| Pendulum Swingup | 560.1 | 376.4 | 5.0 | **839.7** | 806.3 | 0.0 | 0.0 |
| Quadruped Run | 50.5 | 141.5 | 299.7 | **407.0** | 352.3 | 46.2 | **396.6** |
| Quadruped Walk | 49.7 | 123.7 | 107.1 | **660.3** | 352.6 | 55.4 | 445.4 |
| Reacher Easy | 86.5 | 609.3 | 705.8 | **910.2** | **898.9** | 72.7 | 222.4 |
| Reacher Hard | 9.1 | 400.2 | 12.6 | **572.9** | 499.2 | 24.3 | 12.3 |
| Walker Run | 26.9 | 376.2 | 32.7 | 517.1 | **757.8** | 387.2 | 427.6 |
| Walker Stand | 159.3 | 463.5 | 163.8 | **974.1** | **976.7** | 934.8 | 954.8 |
| Walker Walk | 38.9 | 828.8 | 96.0 | 762.9 | **955.8** | 758.0 | 854.7 |
| Median | 78.5 | 431.8 | 101.5 | 734.9 | **808.5** | 31.5 | 475.12 |
| Mean | 225.3 | 504.7 | 211.9 | 677.4 | **739.6** | 214.5 | 442.1 |

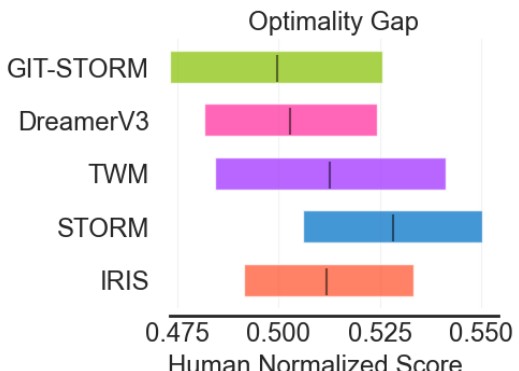

Figure 8: Optimality Gap of the mentioned baselines and GIT-STORM. It shows the amount by each algorithm fails to achieve human performance, where lower values are better.

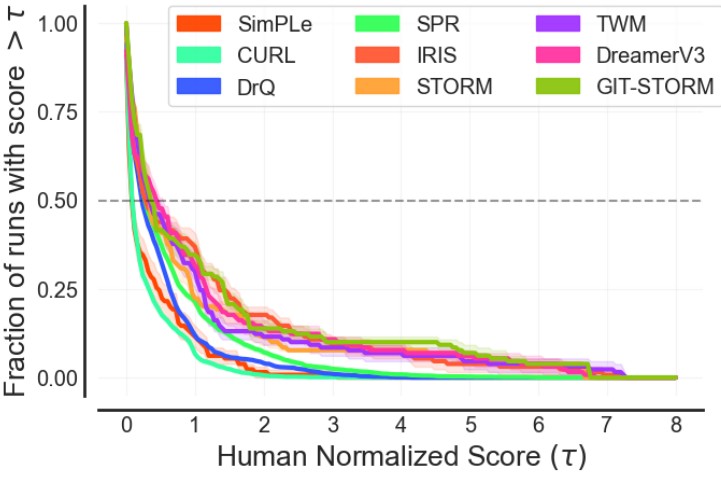

Figure 9: Performance profiles of the considered baseline and our proposed model GIT-STORM. The graph presents the fraction of games above a certain human normalized score.

# D  TRAINING CURVES

In this section, we provide the training curves of GIT-STORM for both Atari 100k and DMC benchmark.

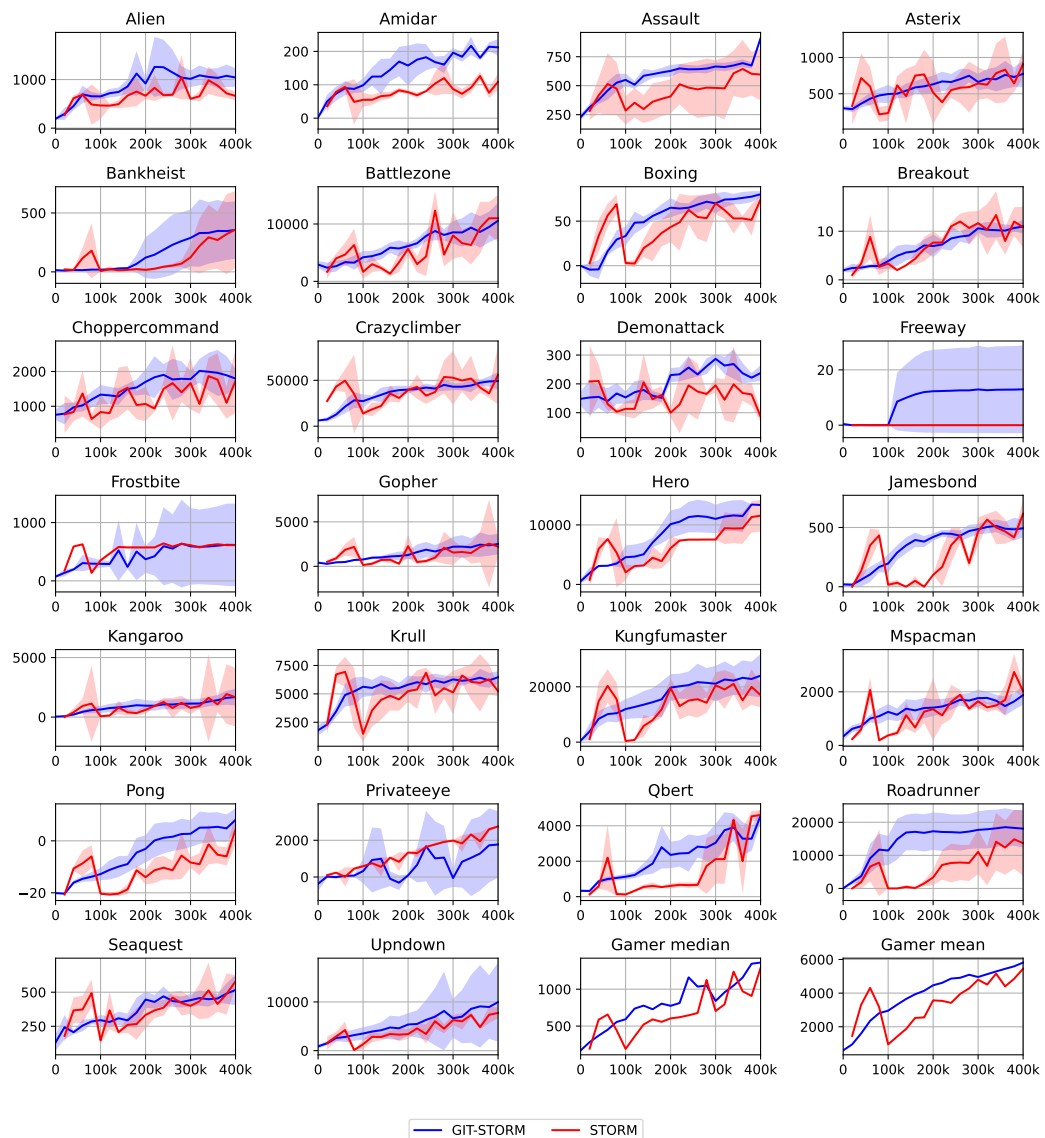

Figure 10: Training profiles across all the checkpoints for the Atari 100k benchmark. The solid line represents the average over 5 seeds while the fill area is defined in terms of maximum and minimum values corresponding to each checkpoint.

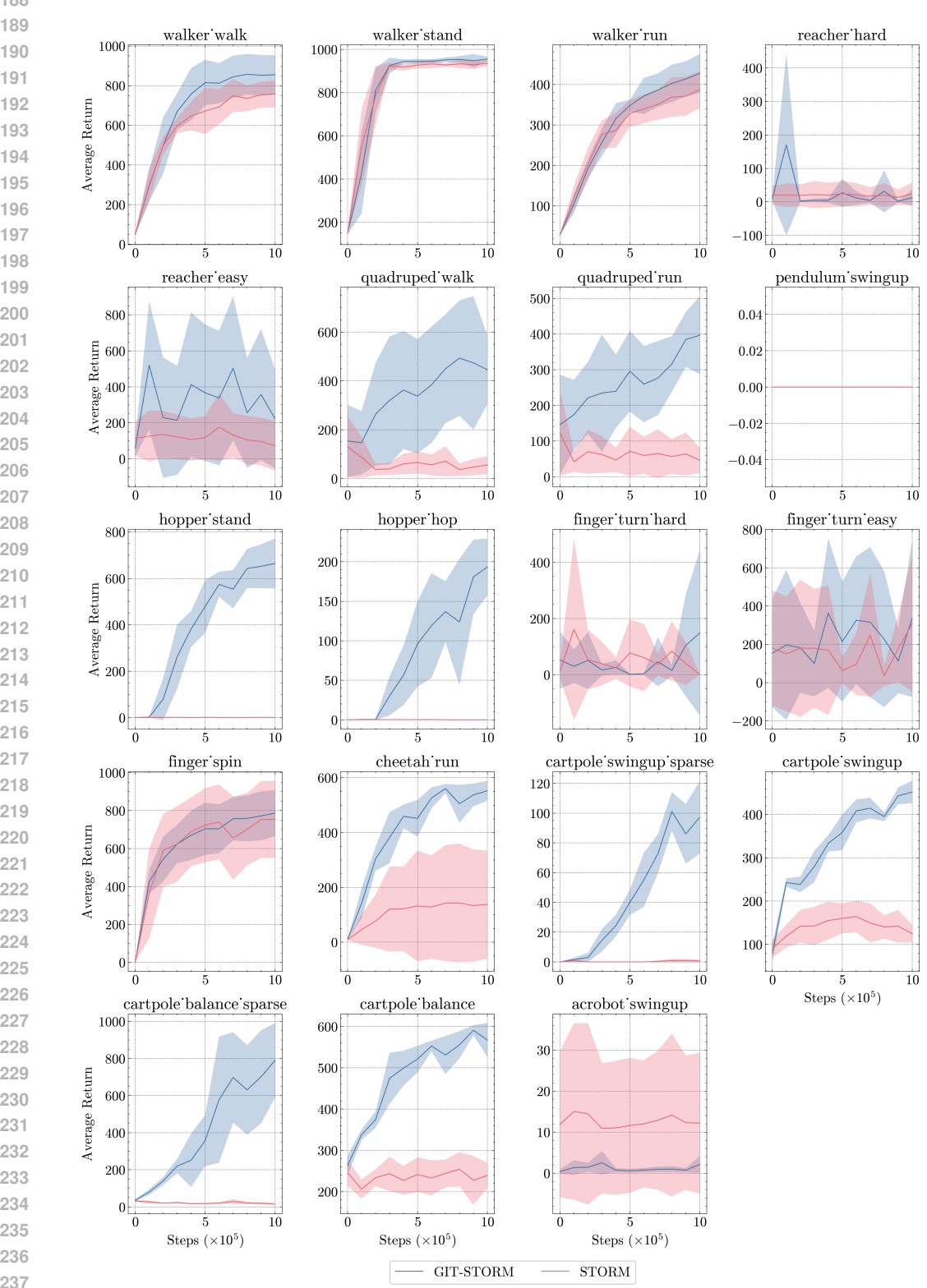

Figure 11: Performance evaluation across all the checkpoints for the DMC benchmark for GIT-STORM and STORM. The solid line represents the average over 5 seeds while the fill area is defined in terms of standard deviation values corresponding to each checkpoint.

# E  ABLATION STUDY

## E.1  GIT-STORM ABLATIONS

In this section, we analyze the contributions of the two primary components that define GIT-STORM:

- **MaskGIT Head**: We compare the performance of the MaskGIT head against a standard MLP head to assess its role in improving downstream results.
- **Logits Computation via Dot Product**: We evaluate the impact of computing logits as the dot product between $\xi_t$ and the MaskGIT embeddings, comparing this approach to the alternative of using an MLP head that takes $\xi_t$ as input and directly outputs logits.

These components are hypothesized to be critical for understanding the capabilities of GIT-STORM and the individual contributions they make to the observed performance improvements.

Figure 12 illustrates an ablation study on three Atari games (Hero, Freeway, and Boxing) and three DMC environments (Walker Walk, Walker Run, and Quadruped Run). Across both sets of environments, the removal of the MaskGIT head consistently results in poorer downstream performance (e.g., lower scores). Additionally, leveraging the dot product between $\xi_t$ and MaskGIT embeddings has a substantial impact in environments such as Freeway, Walker Walk, and Quadruped Run. However, its influence appears negligible in other environments like Hero and Walker Run, suggesting that its efficacy may be context-dependent.

## E.2  DIMENSIONS OF DYNAMIC PRIOR HEAD

In order to find the best configuration for the MaskGIT prior, we conduct experiments on three different environments with different embedding and vocabulary dimensions corresponding to the bidirectional transformer. While the performance of different configurations varies between environments, we find that a bigger embedding size achieves higher scores on average as seen in Figure 13.

As shown in DreamerV3 (Hafner et al., 2023), the model achieves better performance as it increases in the number of trainable parameters. Thus, to provide a fair comparison with STORM, we restrict the transformer corresponding to the MaskGIT prior to a similar number of parameters as the MLP prior defined in STORM.

## E.3  VQ-VAE VS ONE HOT CATEGORICAL

The world model state in model-based RL is represented in terms of a latent representation based on raw observations from the environment. However, there is no clear consensus on the representation of the latent space, with SimPLE (Kaiser et al., 2019) using a Binary-VAE, IRIS (Micheli et al., 2022) using a VQ-VAE while DreamerV3 (Hafner et al., 2023), STORM (Zhang et al., 2023) and TWM (Robine et al., 2023) employ a Categorical-VAE.

While recent methods show empirically the advantages of a Categorical-VAE in Atari environments, there is no comprehensive study on different latent space representations. Thus, Table 7 provides a comparison between a VQ-VAE and Categorical-VAE latent representation in the context of GIT-STORM, motivating our choice of latent space. The comparison is performed on three environments with different levels of complexity in terms of visual representations.

Table 7: Comparison between a VQ-VAE and Categorical-VAE latent representation for the world model state on three Atari 100k environments.

| Game | VQ-VAE | One Hot Categorical |
|---|---|---|
| Boxing | 0 | **81** |
| Hero | 0 | **13351** |
| MsPacman | 255 | **1877** |

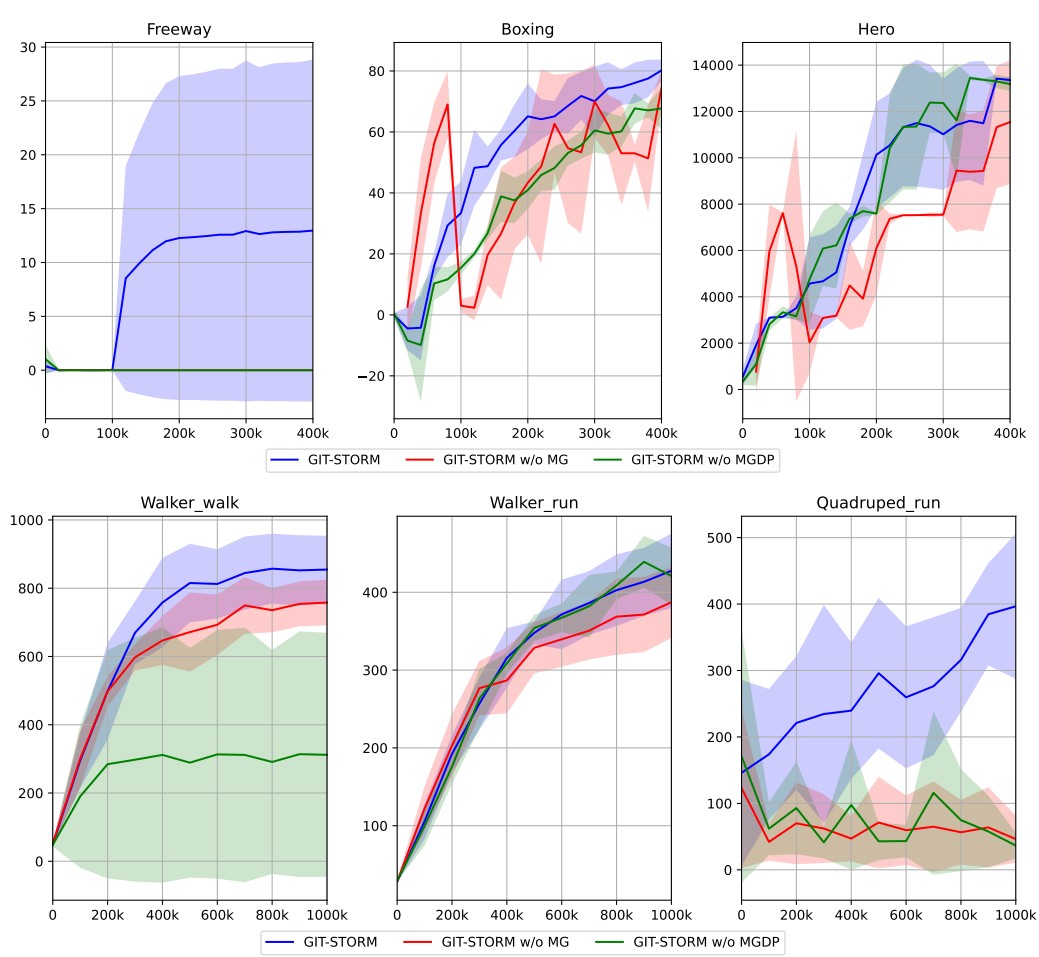

Figure 12: GIT-STORM ablation study on selected Atari and DMC environments: GIT-STORM w/o MG stands for without MaskGIT head, while GIT-STORM w/o MGDP stands for without MaskGIT dot product. All results are averaged across three random seeds.

In order to keep the comparison between the two representations accurate, we scale down the VQ-VAE to only 32 codebook entries, each consisting of 32 dimensions, matching the size of the one-hot categorical representation of 32 categories with 32 classes each. While the VQ-VAE in IRIS (Micheli et al., 2022) uses a considerably bigger vocabulary and embedding size, we believe the additional number of parameters introduced provide a biased estimation of the representation capabilities of the latent space. Moreover, we notice that the VQ-VAE approach introduces a significant overhead in terms of training and sampling time. Table 7 shows that the VQ-VAE latent representations collapse and fail to learn a meaningful policy. In contrast the categorical representation achieves impressive results with the same compute budget.

### E.4 STATE MIXER ANALYSIS

#### E.4.1 STATE MIXER INDUCTIVE BIASES

As described in Sec. 3.1, latent representations $z_t$ and actions $a_t$ are mixed using a state mixer function $g(\cdot)$. To understand the affect of different mixing strategies for the underlying task, we compare three different mixing functions in the DMC benchmark: (1) concatenation, (2) concatenation followed by attention and (3) cross attention between state and actions. Figure 15 illustrates the results. Surprisingly, we find that the simple approach works the best for the tasks – concatenation of state and action significantly outperforms the attention-based approaches in the chosen tasks.

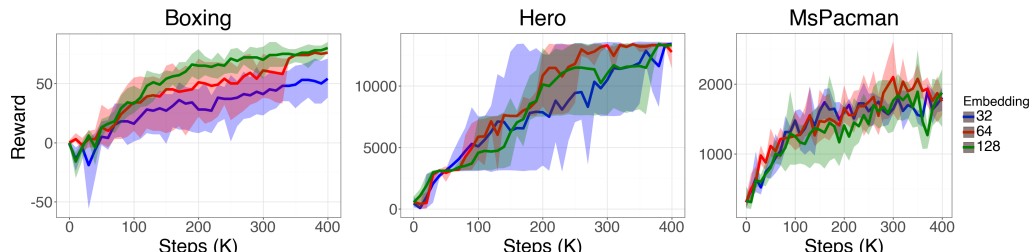

Figure 13: Different MaskGIT configurations for the Bidirectional Transformer embedding size. Bigger embedding sizes achieve better results. Three different seeds were used for this experiment.

### E.4.2 STATE MIXER ABLATIONS

To evaluate the contribution of the State Mixer and its relevance compared to existing approaches, such as iVideoGPT Wu et al. (2024), we conducted an ablation study. This analysis compares the effect of the State Mixer on downstream performance against the approach proposed in iVideoGPT. Figure 14 demonstrates that the State Mixer consistently outperforms the considered baselines. Interestingly, under the given setup, the iVideoGPT approach fails to learn meaningful policies. We hypothesize that this limitation arises from the scale of the training procedure and considered environments. Specifically, iVideoGPT is designed to leverage much larger datasets, enabling it to learn robust representations.

Moreover, we observe that bypassing the State Mixer by directly concatenating and feeding state and action embeddings into the transformer allows the model to learn policies that are meaningful but perform suboptimally compared to the State Mixer-based approach. This finding highlights the effectiveness of the State Mixer in extracting and processing state-action representations crucial for learning optimal policies.

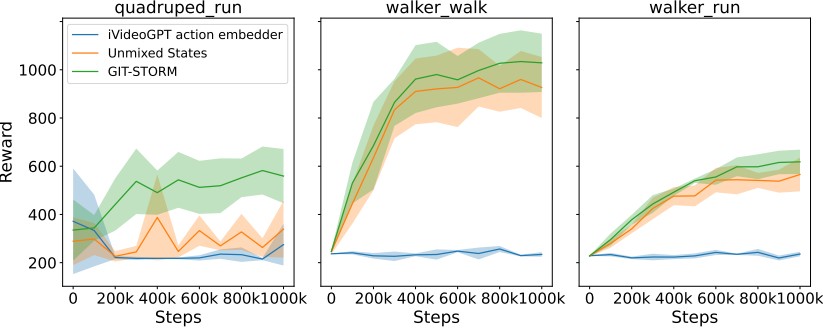

Figure 14: GIT-STORM action embedding approach ablation study on DMC environments. We consider: GIT-STORM, GIT-STORM using iVideoGPT action embedder and GIT-STORM without the State Mixer (labeled as Unmixed States). All results are averaged across three random seeds. GIT-STORM approach consistently outperforms the considered baselines.

## F DYNAMICS HEAD ANALYSIS

### F.1 KL DIVERGENCE COMPARISON

In this section, we present and analyze a comparison between our method and STORM in terms of the KL divergence of the dynamics module. Figure 16 illustrates the KL divergence loss for GIT-STORM and STORM across three environments: Hero, Boxing, and Freeway. It is evident that the KL divergence for GIT-STORM is consistently lower across all three environments, with a particularly significant difference observed in Boxing. This suggests that the dynamics module in

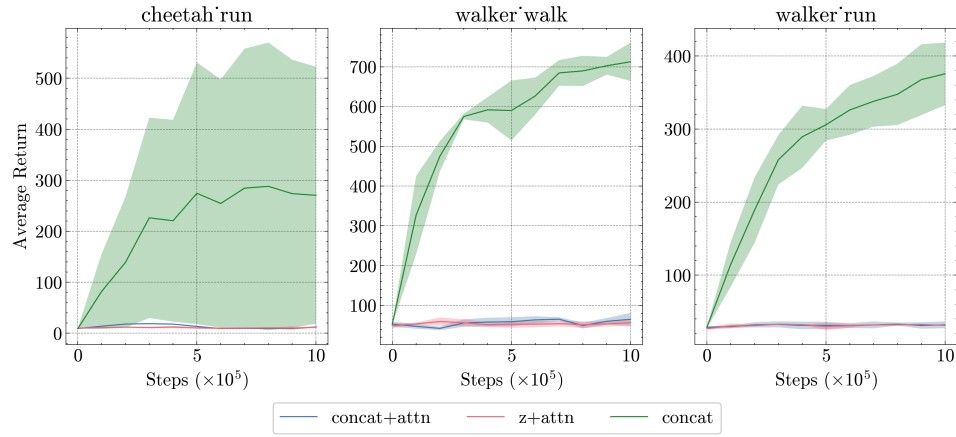

Figure 15: Comparison between different state and action mixing strategies tested in the DMC environments. All results are averaged across three random seeds. We find that simple concatination works the best for the chosen tasks.

GIT-STORM is better equipped to learn state transition dynamics compared to STORM, resulting in more accurate modeling of the underlying system dynamics.

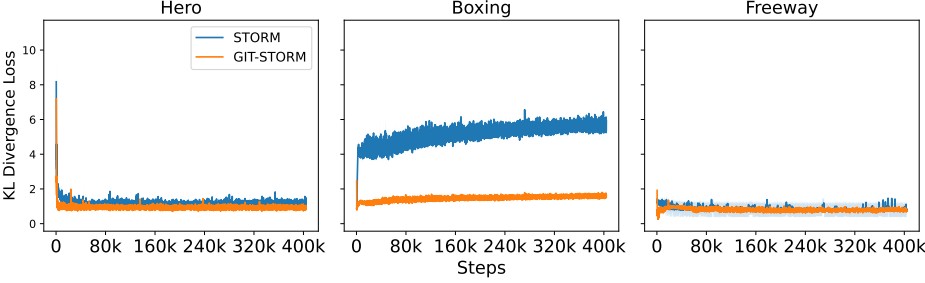

Figure 16: Comparison of GIT-STORM and STORM's KL divergence loss in Hero, Boxing and Freeway. GIT-STORM consistently presents a lower KL divergence. All results are averaged across three random seeds.

### F.2 DYNAMICS HEAD OUTPUT DISTRIBUTION VISUALIZATION

In this section, we inspect the output distributions of the dynamics head generated by the proposed GIT-STORM compared to those produced by STORM. Specifically, Figure 17 illustrates the mean probability distribution for generating a certain token at a given time step and frame. A closer examination of the density functions reveals that the mean distributions typically exhibit two peaks: one near zero, indicating that a given token does not need to be sampled, and a second, smaller peak, representing the confidence level for sampling a specific token.

The higher the second peak and the broader the distribution's support, the more confident the world model is in sampling tokens for a given dynamics state transition. Consistent with the perplexity values presented in Table 4, GIT-STORM produces more refined probability distributions, enabling it to make predictions with greater confidence compared to STORM.

## G VIDEO PREDICTION DOWNSTREAM TASK: TECO

In order to assess the capabilities of the MaskGIT prior in modelling latent dynamics across different tasks, we consider video generation tasks as a representative study. More specifically, we consider Temporally Consistent Transformer for Video Generation (TECO) (Yan et al., 2023) on DeepMind

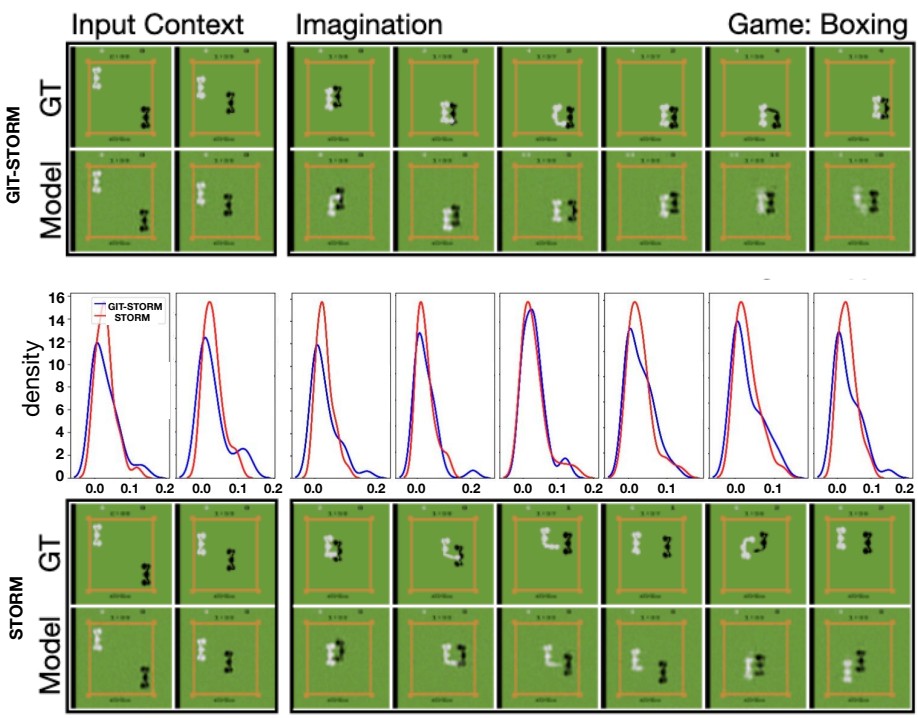

Figure 17: Above: GIT-STORM imagined trajectory in Boxing. Middle: Mean probability distribution of generating a certain token for a given time step and frame. Bottom: STORM imagined trajectory in Boxing.

Lab (DMLab) (Beattie et al., 2016) and Something-Something v.2 (SSv2) (Goyal et al., 2017) datasets. TECO uses a spatial MaskGIT Prior to generate the state corresponding to the next timestep. Table 1 highlights the importance of the prior network and supports our earlier results on the Atari 100k benchmark. Indeed, when replacing the MaskGit prior network with an MLP one with the same number of parameters, the FVD (Unterthiner et al., 2019) on both DMLab and SSv2 datasets significantly increases, going from 48 to 153 and from 199 to 228 in the DMLab and SSv2 datasets respectively.

# H  HYPERPARAMETERS

Table 8: Hyperparameters regarding the dynamics module, training settings and environment. We use the same hyperparameters as STORM (Zhang et al., 2023) to focus our experiments on the MaskGIT prior.

| Hyperparameter | Symbol | Value |
|---|---|---|
| Transformer layers | $K$ | 2 |
| Transformer feature dimension | $D$ | 512 |
| Transformer heads | - | 8 |
| Dropout probability | $p$ | 0.1 |
| World model training batch size | $B_1$ | 16 |
| World model training batch length | $T$ | 64 |
| Imagination batch size | $B_2$ | 1024 |
| Imagination context length | $C$ | 8 |
| Imagination horizon | $L$ | 16 |
| Update world model every env step | - | 1 |
| Update agent every env step | - | 1 |
| Environment context length | - | 16 |
| Gamma | $\gamma$ | 0.985 |
| Lambda | $\lambda$ | 0.95 |
| Entropy coefficiency | $\eta$ | $3 \times 10^{-4}$ |
| Critic EMA decay | $\sigma$ | 0.98 |
| Optimizer | - | Adam (Kingma & Ba, 2014) |
| World model learning rate | - | $1.0 \times 10^{-4}$ |
| World model gradient clipping | - | 1000 |
| Actor-critic learning rate | - | $3.0 \times 10^{-5}$ |
| Actor-critic gradient clipping | - | 100 |
| Gray scale input | - | False |
| Frame stacking | - | False |
| Frame skipping | - | 4 (max over last 2 frames) |
| Use of life information | - | True |
| MaskGIT Transformer layers | - | 4 |
| MaskGIT Transformer feature dimension | - | 128 |
| MaskGIT Transformer heads | - | 8 |
| MaskGIT Dropout probability | - | 0.0 |
| Mask Schedule | - | cosine |
| Draft Rounds | $T_{draft}$ | 1 |
| Revise Rounds | $T_{revise}$ | 1 |
| Repetitions | $M$ | 1 |

Table 9: Specific structure of the image encoder used in GIT-STORM (ours) and STORM (Zhang et al., 2023). The size of the modules is omitted and can be derived from the shape of the tensors. ReLU refers to the rectified linear units used for activation, while Linear represents a fully-connected layer. Flatten and Reshape operations are employed to alter the indexing method of the tensor while preserving the data and their original order. Conv denotes a CNN layer (LeCun et al., 1989), characterized by kernel $= 4$, stride $= 2$, and padding $= 1$. BN denotes the batch normalization layer (Ioffe & Szegedy, 2015).

| Submodule | Output tensor shape |
| --- | --- |
| Input image ($o_t$) | $3 \times 64 \times 64$ |
| Conv1 + BN1 + ReLU | $32 \times 32 \times 32$ |
| Conv2 + BN2 + ReLU | $64 \times 16 \times 16$ |
| Conv3 + BN3 + ReLU | $128 \times 8 \times 8$ |
| Conv4 + BN4 + ReLU | $256 \times 4 \times 4$ |
| Flatten | 4096 |
| Linear | 1024 |
| Reshape (produce $\mathcal{Z}_t$) | $32 \times 32$ |

Table 10: Structure of the image decoder. DeConv denotes a transpose CNN layer (Zeiler et al., 2010), characterized by kernel $= 4$, stride $= 2$, and padding $= 1$.

| Submodule | Output tensor shape |
| --- | --- |
| Random sample ($z_t$) | $32 \times 32$ |
| Flatten | 1024 |
| Linear + BN0 + ReLU | 4096 |
| Reshape | $256 \times 4 \times 4$ |
| DeConv1 + BN1 + ReLU | $128 \times 8 \times 8$ |
| DeConv2 + BN2 + ReLU | $64 \times 16 \times 16$ |
| DeConv3 + BN3 + ReLU | $32 \times 32 \times 32$ |
| DeConv4 (produce $\hat{o}_t$) | $3 \times 64 \times 64$ |

Table 11: Action mixer $\zeta_t = g_\theta(z_t, a_t)$. Concatenate denotes combining the last dimension of two tensors and merging them into one new tensor. The variable $A$ represents the action dimension, which ranges from 3 to 18 across different games. $D$ denotes the feature dimension of the Transformer. LN is an abbreviation for layer normalization (Ba et al., 2016).

| Submodule | Output tensor shape |
| --- | --- |
| Random sample ($z_t$), Action ($a_t$) | $32 \times 32$, $A$ |
| Reshape and concatenate | $1024 + A$ |
| Linear1 + LN1 + ReLU | $D$ |
| Linear2 + LN2 (output $e_t$) | $D$ |

Table 12: Positional encoding module. $w_{1:T}$ is a learnable parameter matrix with shape $T \times D$, and $T$ refers to the sequence length.

| Submodule | Output tensor shape |
| --- | --- |
| Input ($e_{1:T}$) | |
| Add ($e_{1:T} + w_{1:T}$) | $T \times D$ |
| LN | |

Table 13: Transformer block. Dropout mechanism (Srivastava et al., 2014) can prevent overfitting.

| Submodule | Module alias | Output tensor shape |
|---|---|---|
| Input features (label as $x_1$) | | $T \times D$ |
| Multi-head self attention
Linear1 + Dropout($p$)
Residual (add $x_1$)
LN1 (label as $x_2$) | MHSA | $T \times D$ |
| Linear2 + ReLU
Linear3 + Dropout($p$)
Residual (add $x_2$)
LN2 | FFN | $T \times 2D$
$T \times D$
$T \times D$
$T \times D$ |

Table 14: MLP settings. A 1-layer MLP corresponds to a fully-connected layer. 255 is the size of the bucket of symlog two-hot loss (Hafner et al., 2023).

| Module name | Symbol | MLP layers | Input/ MLP hidden/ Output dimension |
|---|---|---|---|
| Reward head | $p_\phi$ | 3 | D/ D/ 255 |
| Termination head | $p_\phi$ | 3 | D/ D/ 1 |
| Policy network | $\pi_\theta(a_t|s_t)$ | 3 | D/ D/ A |
| Critic network | $V_\psi(s_t)$ | 3 | D/ D/ 255 |

## I  COMPUTATIONAL RESOURCES

Throughout our experiments, we make use of NVIDIA A100 and H100 GPUs for both training and evaluation on an internal cluster, a summary of which can be found in Table 15. For the Atari 100k benchmark, we find that each individual experiment requires around 20 hours to train. For the video prediction tasks, DMLab requires 3 days of training on 4 NVIDIA A100 GPUs. For DMC Vision tasks, we used H100 GPUs to sample from 16 environments concurrently, which reduced our training time to only 8 hours for 1M steps. Compared to this, using A100 for one environment takes 7 days. We acknowledge that the research project required more computing resources than the reported ones, due to preliminary experiments and model development.

Table 15: Summary of resources used in experiments.

| Experiment type | GPU Type | # of Days to train |
|---|---|---|
| Atari100k | 1x A100 | 20 hours |
| DMLab | 4x A100 | 3 days |
| DMC Vision | 1x A100 | 8 hours |

## J  BASELINES

To assess our approach downstream capabilities on Atari 100k we select the following baselines: SimPLe (Kaiser et al., 2019) trains a policy using PPO (Schulman et al., 2017) leveraging a world model represented as an action-conditioned video generation model; TWM (Robine et al., 2023) uses a transformer-based world model that leverages a Transformer-XL architecture and a replay buffer which uses a balanced sampling scheme (Dai et al., 2019). IRIS (Micheli et al., 2022), that uses a VideoGPT (Yan et al., 2021) based world model; DreamerV3 (Hafner et al., 2023), a general algorithm which achieves SOTA results on a multitude of RL benchmarks. Lastly, we consider STORM (Zhang et al., 2023), an efficient algorithm based on DreamerV3 that uses the transformer architecture for the world model. Since Hafner et al. (2023) shows that the replay buffer size is a scaling factor, to present a fair comparison we reproduce DreamerV3, which uses a replay buffer of 1M samples by default and full precision variables for the Atari 100k benchmark, using a replay buffer of $100K$ samples and half precision variables, consistent with our approach. Moreover, since STORM does not follow the evaluation protocol proposed in Agarwal et al. (2021), after setting reproducible seeds, we reproduce STORM on the Atari 100k benchmark using the code released by the authors, and report the results as a result of running the released code.

For DMC Suite, we consider several state-of-the-art algorithms. Soft Actor-Critic (SAC) (Haarnoja et al., 2018) is a popular algorithm for continuous control tasks, known for its data efficiency due to the use of experience replay. However, SAC often requires careful tuning, particularly for the entropy coefficient, and its performance can degrade when handling high-dimensional input spaces (Hafner, 2022). Another baseline is Proximal Policy Optimization (PPO) (Schulman et al., 2017), a widely-used RL algorithm recognized for its robustness and stability across a range of tasks. Additionally, we include DrQ-v2 (Yarats et al., 2021) and CURL (Laskin et al., 2020), both of which are tailored for visual environments. These methods leverage data augmentation to improve the robustness of learned policies, making them highly effective in scenarios where pixel-based observations dominate. Finally, we consider DreamerV3, which is the current state-of-the-art in this environment.

# K  METRICS

In order to meaninfgully evaluate the considered baselines we follow the protocol suggested in Agarwal et al. (2021), which proposes the following metrics for a statistically grounded comparison:

- **Human Normalized Score**: To account for the discrepancies between raw score ranges in Atari games, and at the same time comparing the algorithm's capabilities with the human benchmark, the human normalized score is used to assess the performance of an algorithm on a specific environment. The Human Normalized Score is defined as $\frac{agent_{score} - random_{score}}{human_{score} - random_{score}}$.

- **Human Mean**: The Human Mean is an aggregate metric used to assess the performance across the whole Atari benchmark. The mean is computed using the Human Normalized Score for each environment, as previously defined.

- **Human Median**: Similar to the Human Mean, the Human Median is an aggregate metric across the Atari benchmark that is insensitive to high-score environments, which instead harm the statistical significance of the Human mean. According to Agarwal et al. (2021), both the Human Mean and Human Median are necessary to assess the performance of an algorithm in Atari.

- **Interquantile Mean (IQM)**: Interquantile Mean is a popular statistical tool that only considers $50\%$ of the results, effectively ignoring the lowest and highest performing environments. IQM aims to address the shortcomings of the Human Mean by ignoring outliers, while being more statistically significant than the Human Median, which only considers a single value.

- **Performance profiles (score distributions)**: Considering the variety of score ranges across different Atari environments, some of which may be heavy-tailed or contain outliers, point or interval estimates provide an incomplete picture with respect to an algorithm's performance. Performance profiles aim to alleviate this issues by revealing performance variability across tasks more significantly than interval and point estimates, like the Human Mean and Human Median.

- **Optimality Gap**: The Optimality Gap represents another alternative to the Human Mean, and accounts for how much the algorithm fails to meet a minimum Human Normalized Score of $\gamma = 1$. The metric considers $\gamma$ as the desirable target and does not account for values greater than it. In the context of the Atari benchmark, $\gamma = 1$ represents the human performance. Using the Optimality Gap, the algorithms are compared without taking in consideration super-human performance, which is considered irrelevant.

- **Probability of Improvement**: Instead of treating algorithm's comparison as a binary decision (better or worse), the Probability of Improvement, indicates a probability corresponding to how likely it is for algorithm $X$ to outperform algorithm $Y$ on a specific task.

For sequence modelling and video prediction task, we use the following metrics:

- **Perplexity**: Perplexity is mathematically defined as the exponentiated average negative log-likelihood of a sequence. Given a sequence of categorical representations $z_0, z_1, \ldots, z_t$, the perplexity of $z$ is computed as:

$$\text{PPL}(z) = \exp\left\{ -\frac{1}{t} \sum_{i=1}^{t} \log p_\phi(z_i \mid z_{<i}) \right\}.$$

Here, $\log p_\theta(z_i \mid z_{<i})$ is the log-likelihood of the $i$-th token, conditioned on the preceding tokens $z_{<i}$, according to the model. In this context, perplexity serves as a measure of the model's ability to predict the tokenized representations of images in a sequence.

- **Fréchet Video Distance (FVD)**: Introduced in (Unterthiner et al., 2019), FVD is a metric designed to evaluate the quality of video generation models. It builds on the idea of the widely-used Fréchet Inception Distance (FID), which is applied to assess the quality of generated images, but extends it to video by incorporating temporal dynamics. FVD is particularly effective for comparing the realism of generated videos with real video data, making it a crucial metric in video prediction and generation tasks.

