# OpenReview forum: "Masked Generative Priors Improve World Models Sequence Modelling Capabilities"
_ICLR.cc/2025/Conference — Submitted to ICLR 2025_

### Official Review · Reviewer_JuLK · 2024-10-22

**Soundness:** 2
**Presentation:** 2
**Contribution:** 3
**Rating:** 5
**Confidence:** 4

**Summary:**

This paper proposes GIT-STORM, which utilizes a MaskGIT model instead of an MLP for the prior head in world models (based on STORM). It also makes minor modifications (a state mixer) to support continuous actions. Experiments are done on Atari100k and DMC benchmarks, considering both policy learning and video prediction performance. GIT-STORM outperforms its base method STORM.

**Strengths:**

To my knowledge, MaskGIT models, with their strong expressiveness, are not yet utilized for world models in the MBRL community.

**Weaknesses:**

1. The illustration and descriptions of the model are confusing. Can authors provide more insights for their specific designs?
   - In Figure 1 (left), it seems that GIT-STORM uses masked $z_t$ as inputs for reconstructing $z_{t+1}$. This is strange since, in the original MaskGIT, we mask and reconstruct masked target tokens. Similarly, I think it is more reasonable to mask $z_{t+1}$ as inputs.
   - In Figure 1 (left), there is no $\xi_t$ but only $\eta_t$.
   - Also, the dot product seems to be a commonly used trick that ties weights for embedding and linear layer before Softmax. If so, relevant literature should be cited.
   - The Draft-and-Revise decoding scheme, if not proposed by this work, should be moved into a preliminary section.
2. The contribution to supporting continuous actions is overclaimed (as 'for the first time'). In fact, concatenating or summating continuous inputs with hidden states is a too straightforward approach in current VLA models (e.g., OpenVLA for inputting continuous visual representations) and action-conditioned video prediction models (e.g., iVideoGPT for inputting continuous actions).
3. The performance of GIT-STORM on DMC is outperformed by its base method, DreamerV3.

**Questions:**

There are also some minor questions:

1. In Line 309, why KV cache can improve sample efficiency? Do you mean computational efficiency?
2. To my knowledge, perplexity is a metric whose lower values mean better. However, in Table 3, higher perplexity is marked as better.
3. In Figure 6, the quadruped agents are too small in the images. This work seems to have used an unusual camera setting for these tasks.

If the authors well address my concerns, I am willing to improve my rating.

---

> ### Author Response · Authors · 2024-11-23
> **Q&A 1**
>
> We thank the reviewer for such insightful and important questions. We are really happy to answer and fulfil any remaining doubts. Here we list aggregated questions and the related answers.
>
> Q1: The illustration and descriptions of the model are confusing. Can authors provide more insights for their specific designs?
> In Figure 1 (left), it seems that GIT-STORM uses masked as inputs for reconstructing . This is strange since, in the original MaskGIT, we mask and reconstruct masked target tokens. Similarly, I think it is more reasonable to mask as inputs.
> In Figure 1 (left), there is no but only .
> Also, the dot product seems to be a commonly used trick that ties weights for embedding and linear layer before Softmax. If so, relevant literature should be cited.
> The Draft-and-Revise decoding scheme, if not proposed by this work, should be moved into a preliminary section.
>
>
> A1: We thank the reviewer for such detailed feedback. For our design, we based on the latent MaskGIT implementation proposed in TECO[1]. The original MaskGIT masks and reconstructs target tokens because they do not deal with state transitions but image reconstruction/generation, which does not involve dynamics. In our case, we want to generate tokens of the next time step, or in other words, we want to predict the dynamics state transition of $z_t$. Therefore, as proposed in TECO, in order to predict $z_{t+1}$  we use as input $z_t$, as context, and $h_t$, which is supposed to contain information about the next time step.
>
> We added $\xi_t$ to the model architecture figure.
>
> We could not find any specific relevant model that uses the dot product operation as in GIT-STORM. If there is any paper that is relevant to our work, please give us the references and we will add the citations.
>
> We agree with the reviewer that this should be the case. However, for the sake of readability, we decided to present it in the methodology section and specify that we are not proposing it for the first time.

---

> ### Author Response · Authors · 2024-11-23
> **Q&A 2**
>
> Q2: The contribution to supporting continuous actions is overclaimed (as 'for the first time'). In fact, concatenating or summating continuous inputs with hidden states is a too straightforward approach in current VLA models (e.g., OpenVLA for inputting continuous visual representations) and action-conditioned video prediction models (e.g., iVideoGPT for inputting continuous actions).
>
>
> A2: We thank the reviewer for this feedback. Current VLA models (e.g., OpenVLA) do not use actions within their input space, instead they output actions. This is a very important difference since the model does not need to learn a causality relationship between states and actions. In other words, VLA models can be interpreted as policies and not world models, which makes them different modules and therefore cannot be compared to our model. When it comes to action-conditioned video prediction models (e.g., iVideoGPT), they do not mix actions and states, they use embeddings to map actions and input them as separated entities together with the states.
>
> We found this question particularly interesting and we decided to perform an experiment to compare the action embedder approach used in iVideoGPT against our action state mixer. Moreover, we also include a version where we simply concatenate actions and states, without mixing them, to validate whether the contribution to the performance of our approach comes from the mixer or from concatenating actions and states. This analysis compares the effect of the State Mixer on downstream performance against the approach proposed in iVideoGPT. Figure 14 demonstrates that the State Mixer consistently outperforms the considered baselines.
> Interestingly, under the given setup, the iVideoGPT approach fails to learn meaningful policies. We hypothesize that this limitation arises from the scale of the training procedure and considered environments. Specifically, iVideoGPT is designed to leverage much larger datasets, enabling it to learn robust representations. Moreover, we observe that bypassing the State Mixer by directly concatenating and feeding state and action embeddings into the transformer allows the model to learn policies that are meaningful
> but perform suboptimally compared to the State Mixer-based approach. This finding highlights the effectiveness of the State Mixer in extracting and processing state-action representations crucial for learning optimal policies.
>
> Finally, we would like to remark that combining actions and states meaningfully is not a straightforward problem, especially when it comes to continuous actions and discrete states. Therefore, we believe the community would greatly benefit from our work, as it explicitly shows and highlights an approach to do that, while all the other cited papers do not expand on this problem and simply state what they used.

---

> ### Author Response · Authors · 2024-11-23
> **Q&A 3 and 4**
>
> Q3: The performance of GIT-STORM on DMC is outperformed by its base method, DreamerV3.
>
> We thank the reviewer for the feedback. We are aware that DreamerV3 beats GIT-STORM on the DMC benchmark. However, we believe that this does not imply that the contribution of our work is meaningless or not good enough. Indeed, the message that we want to bring to our community is that masked generative priors increase the sequence modeling capabilities of transformer-based world models. This is quite clear when looking at the comparison between STORM and GIT-STORM. Moreover, this is also the very first work where transformer-based world models are benchmarked on the whole DMC suite. The main reason is that using continuous actions and discrete representations is not trivial and very difficult to perform meaningfully.
>
> Q4: In Line 309, why KV cache can improve sample efficiency? Do you mean computational efficiency?
>
> The KV cache improves computational efficiency by storing key-value pairs from previously computed attention operations, eliminating redundant calculations, and reducing the computational cost of sequence modeling in transformers. In the model-based reinforcement learning context, this enhanced computational efficiency translates directly to improved sample efficiency, as the primary bottleneck in RL training is the number of sampled trajectories required to train policies. By enabling the model to process more trajectories within the same computational budget, the KV cache accelerates training, reduces the need for environment interactions, and facilitates longer sequence modeling without the quadratic cost scaling of traditional attention mechanisms. This dual benefit of faster training and efficient resource utilization makes the KV cache an important component for improving sample efficiency.

---

> ### Author Response · Authors · 2024-11-23
> **Q&A 5 and 6**
>
> Q5: To my knowledge, perplexity is a metric whose lower values mean better. However, in Table 3, higher perplexity is marked as better.
>
> A5: In the context of computer vision and Autoencoder networks that learn image representations perplexity is maximized to encourage uniform utilization of codebook entries in models like Vector Quantized Variational Autoencoders (VQ-VAEs).
> A higher perplexity indicates that the codebook entries are used more uniformly, which is desirable because it promotes diversity and richness in representation. Maximizing perplexity prevents codebook collapse, where only a few codes are used, leading to loss of information and reduced model expressiveness. It also improves reconstruction quality by allowing the model to capture more complex and diverse image features, enhancing detail preservation in reconstructed images.
>
> Here you can find some other papers where higher perplexity is considered better:
>
> [2] Yuhta Takida et al. SQ-VAE: Variational Bayes on Discrete Representation with Self-annealed Stochastic Quantization, 2022. https://arxiv.org/pdf/2205.07547
>
> [3] Minyoung Huh et al. Straightening Out the Straight-Through Estimator: Overcoming Optimization Challenges in Vector Quantized Networks, 2023. https://proceedings.mlr.press/v202/huh23a/huh23a.pdf
>
> Q6: In Figure 6, the quadruped agents are too small in the images. This work seems to have used an unusual camera setting for these tasks.
>
> A6: We thank the reviewer for the feedback. We used the standard DMC wrapper provided by torch RL: https://pytorch.org/rl/stable/reference/generated/torchrl.envs.DMControlWrapper.html
>
> When it comes to visualizing the results, usually the images are upsampled for visualization purposes. We did not have the time to do that before the submission, we will try to fix it before the rebuttal period ends. For now, we removed the sequence from the paper.

---

> > ### Comment · Reviewer_JuLK · 2024-11-25
> >
> > I am grateful to the authors for the detailed responses they have provided. Some of my concerns have been addressed perfectly, but I still have the following questions that need clarification.
> >
> > - A1: I partially understand that the MaskGit head in your work is used to simulate z transitions. However, I find the sampling of this transition very unclear. Could the author elaborate on which z is actually z_t and which is actually z_{t+1} in Algorithm 1? It seems that we need to input masked z_t to the header in order to output unmasked z_{t+1}, but in Algorithm 1, we are only iteratively decoding a single z.
> > - A1: Moreover, the dot product operation seems exactly the weight tying strategy widely used in language models and MaskGiT implementations. See the reference links below:
> >   - GPT2: https://github.com/huggingface/transformers/issues/1993; https://datascience.stackexchange.com/questions/123149/why-do-gpt-models-use-a-transpose-of-the-embedding-matrix-to-convert-outputs-to
> >   - MaskGIT: https://github.com/dome272/MaskGIT-pytorch/blob/cff485ad3a14b6ed5f3aa966e045ea2bc8c68ad8/bidirectional_transformer.py#L133; https://github.com/valeoai/Maskgit-pytorch/blob/b0b2b3cc11cffd0b159f22dc1c6e73a7e8b53db3/Network/transformer.py#L189
> > - A4: Do you mean that due to improved computational efficiency, you can set a larger update-to-data ratio (related to Update agent every env step in Table 8) thus improve the sample efficiency of model-based RL? Can you provide this hyperparameter of GIT-STORM and baselines for a direct comparison? For example, is the sampling efficiency improved because GIT-STORM has a larger "Update agent every env step"? If not, then the KV cache does not seem to contribute to sampling efficiency.

---

> > > ### Author Response · Authors · 2024-11-25
> > > **Answers**
> > >
> > > We thank the reviewer for the feedback, we are happy to answer any further questions!
> > >
> > > Q1: I partially understand that the MaskGit head in your work is used to simulate z transitions. However, I find the sampling of this transition very unclear. Could the author elaborate on which z is actually z_t and which is actually z_{t+1} in Algorithm 1? It seems that we need to input masked z_t to the header in order to output unmasked z_{t+1}, but in Algorithm 1, we are only iteratively decoding a single z.
> > >
> > > A1: We now understand the source of confusion and will try to answer as clearly as possible.
> > >
> > > To answer your question, we would like to give a high-level overview of the proposed method during training and inference time. Indeed, the Draft-and-Revise decoding scheme (Algorithm 1) is used only during inference (i.e., imagination and test phases).
> > >
> > > Training Phase:  As shown in Fig. 7, latent representations $z_t$ are first extracted from observation $o_t$. Then, $z_t$ is mixed to $a_t$ and given as input to the autoregressive transformer, as shown in Fig. 1. The Transformer output $h_t$ is then given as input to the MaskGIT head, together with a masked version of $z_t$. Finally, the MaskGIT head returns $z_{t+1}$.
> > >
> > > Imagination/Test phase:
> > > In this case, the Draft-and-Revise scheme is used:  The Transformer output $h_t$ is given as input to the MaskGIT head, together with an empty version of $z_t$, which only contains masks ( referenced as $z^{\text{empty}}$  in Algorithm 1).
> > >
> > > When it comes to Algorithm 1, we are describing the sampling operation of the MaskGIT head that is, therefore $z_t$ is the $z$ that it takes as input, and $z_{t+1}$ is the one that is outputted.  Then, the Draft-and-Revise scheme is executed as described in Algorithm 1, with a first draft phase and N revision phases. In each phase, the MaskGIT head is used to compute either draft or revision logits, the only difference with training time is that this time we do not use $z_t$. Instead, following TECO, we start from a fully masked $z_t$ to avoid the model being biased from the previous state and condition the sampling of the next state dynamics transition only on $h_t$. As a result, once the revision phase has been executed, we get $z_{t+1}$ as output.
> > >
> > > Going back to your question: "Could the author elaborate on which z is actually z_t and which is actually z_{t+1} in Algorithm 1?", it should be clear now that $z_t$ is never used and $z_{t+1}$ is the output of Algorithm 1.
> > >
> > > We thank the reviewer again for this question, as it made us realize that the algorithm description, which was meant to be as general, is probably unclear as of now. To address this, we revised the algorithm description to be more integrated with the paper and now the usage of the MaskGIT head is expressed explicitly.
> > >
> > > Q2: Moreover, the dot product operation seems exactly the weight tying strategy widely used in language models and MaskGiT implementations. See the reference links below:
> > > GPT2: https://github.com/huggingface/transformers/issues/1993; https://datascience.stackexchange.com/questions/123149/why-do-gpt-models-use-a-transpose-of-the-embedding-matrix-to-convert-outputs-to
> > > MaskGIT: https://github.com/dome272/MaskGIT-pytorch/blob/cff485ad3a14b6ed5f3aa966e045ea2bc8c68ad8/bidirectional_transformer.py#L133; https://github.com/valeoai/Maskgit-pytorch/blob/b0b2b3cc11cffd0b159f22dc1c6e73a7e8b53db3/Network/transformer.py#L189
> > >
> > > A2: We thank the reviewer for indicating some references about weight tying strategy. We apologize for not noticing before that indeed such an approach has been formalized and used in several known architectures, including MaskGIT and GPT models. We updated the text that describes the dot product operation and added references to the papers you mentioned and the one that formalizes the weight tying strategy.

---

> > > > ### Comment · Reviewer_JuLK · 2024-11-29
> > > >
> > > > I appreciate the clarification from the authors. However, I still do not fully understand.
> > > >
> > > > - Q1: I get that $z_t$ is not never used in Algorithm 1. But let's look at line 6 of Algorithm 1. It seems that the output (corresponding to $z_{t+1}$) is also used as input to MaskGIT for multiple iterations. This creates a gap between training and imagining/testing, because during training, the MaskGIT input is $z_t$.
> > > >
> > > > - Q2: When we talk about a higher sample efficiency of the MBRL algorithm, we mean that the algorithm can achieve higher task performance with the same or fewer environment steps (rather than training time; see https://ai.stackexchange.com/a/5295). Therefore, the author's statement that "quicker the generation (or sampling) of imagined trajectories and, hence, the higher the sample efficiency" is not true, since the generation speed is independent of the task performance for the same number of environment steps.

---

> > > > > ### Author Response · Authors · 2024-11-29
> > > > >
> > > > > We thank the reviewer for replying, we are really happy to answer your questions.
> > > > >
> > > > > Q1: I get that $z_t$ is not never used in Algorithm 1. But let's look at line 6 of Algorithm 1. It seems that the output (corresponding to $z_{t+1}$) is also used as input to MaskGIT for multiple iterations. This creates a gap between training and imagining/testing, because during training, the MaskGIT input is $z_t$.
> > > > >
> > > > > A1: We understand that this concept may be a bit confusing, so we try to expand it a bit more here. Let's break down Training and Imagination/Test; during training, the MaskGIT input is a **masked** version of $z_t$ and not the full $z_t$. Since the masking operation is done randomly, we can basically rewrite the step as in line 6 of Algorithm 1, which not only takes z as input but also $\Pi$ and h_t. Using different iterations does not create a gap between training and imagination\test, because essentially we are doing the same operation N times, where the only things that change are z and $\Pi$. But this is not different from what happens during training, since per every step the MaskGIT head takes a different z which is randomly masked (e.g., $\Pi$).
> > > > >
> > > > > Q2: When we talk about a higher sample efficiency of the MBRL algorithm, we mean that the algorithm can achieve higher task performance with the same or fewer environment steps (rather than training time; see https://ai.stackexchange.com/a/5295). Therefore, the author's statement that "quicker the generation (or sampling) of imagined trajectories and, hence, the higher the sample efficiency" is not true, since the generation speed is independent of the task performance for the same number of environment steps.
> > > > >
> > > > > A2: We thank the reviewer for such an insightful question. The StackExchange website you shared is related to the paper called "Sample Efficient Actor-Critic Witt Experience Replay. https://arxiv.org/pdf/1611.01224 .
> > > > >
> > > > > The paper states the following: "In particular, every time an agent acts upon the environment, an expensive simulation
> > > > > step is conducted. Thus to reduce the cost of simulation, we need to reduce the number of simulation
> > > > > steps (i.e. samples of the environment). This need for sample efficiency is even more compelling
> > > > > when agents are deployed in the real world.". We agree that this sentence is in accordance with what you wrote as well -  "with sample efficiency, we mean that the algorithm can achieve higher task performance with the same or fewer environment steps".
> > > > >
> > > > > We apologize for making a wrong statement, the confusion came from the fact that all models use the same number of environment steps, for instance, Atari100k is called like this because every given algorithm does 100k interactions with a given Atari environment. Considering that our model performs the best, according to many statistical metrics (i.e., Probability of improvement, IQM, mean, Optimality GAP), and that every model uses the same number of environment steps, according to the definition you gave to sample efficiency, our model should be the most sample efficient. However, within the context of the question you asked before - whether KV caching improves this sample efficiency or not - we agree with you that this is not the case. KV caching improves the computational efficiency of imagined trajectory generation as you suggested. We thank you for your valuable feedback! We corrected the statement we made in the previous answer.
> > > > >
> > > > > Please let us know if you have any other feedback, they are very useful!

---

> > > > > > ### Comment · Reviewer_JuLK · 2024-11-29
> > > > > >
> > > > > > Q1 (continued): As stated by the authors, during training, the MaskGIT input is a masked version of $z_t$. Based on this statement, I want the authors to provide direct answers to the following subquestion:
> > > > > > - SQ1: In Line 3 of Algo 1, should $z^0$ be a draft of $z_{t+1}$?
> > > > > > - SQ2: If the above is true, then in Line 3&4 of Algo 2 (called from Line 6 of Algo 1 with $\gamma=1$), is the input of the MaskGIT's BidirectionalTransformer a masked version of $z^0$, namely a masked version of $z_{t+1}$?
> > > > > > - SQ3: If the above is true, then we face a gap between training and sampling, because for the same desired outputs $z_{t+1}$, our inputs is masked $z_t$ during training but masked $z_{t+1}$ during sampling.

---

> > > > > > > ### Author Response · Authors · 2024-11-30
> > > > > > >
> > > > > > > We thank the reviewer for the question. We are happy to expand on the MaskGIT prior and its usage in Algorithms 1 and 2.
> > > > > > >
> > > > > > >
> > > > > > > SQ1: In Line 3 of Algo 1, should $z^0$ be a draft of $z_{t+1}$?
> > > > > > > A1: Not exactly. The MaskGIT prior respects the following equation: $z_{t+1} \sim MaskGIT Head(z_t, h_t)$, therefore when drafting $z^0 \sim MaskGIT Head( fully masked (z_t) , h_t)$, $z^0$ is called draft but is not necessarily and literally a draft of $z_{t+1}$.
> > > > > > >
> > > > > > > SQ3: If the above is true, then we face a gap between training and sampling, because for the same desired outputs $z_{t+1}$, our inputs is masked  $z_t$ during training but masked $z_{t+1}$ during sampling.
> > > > > > >
> > > > > > > A3: We are aware that this regime is not exactly as in training, it works well and could be explained considering that often there is not a big difference from one frame to another one, therefore we can expect that some of the tokens in $z_t$ remain the same in $z_{t+1}$. If we now consider a masked version of $z_t$ and a masked version of $z_{t+1}$, if some of the few tokens that have changed get masked the two representations would be even more similar. Although we are not sure if there is a way to prove this, we can say that they may appear as sampled from the same distribution and that, therefore, using a masked version of $z_{t+1}$ would be considered an in-distribution regime since we always use a masked version of $z_t$ during training.
> > > > > > >
> > > > > > > Although this is just a potential explanation of why it works, our explanation is also corroborated by the experimental and downstream results, which confirm and show that the model works. Finally, we are not the first ones that use this approach, indeed TECO[1] and the Draft-and-Revise algorithm [2] paper use the same approach in their proposed architectures, showing that this approach works well even there.
> > > > > > >
> > > > > > > Moreover, to answer reviewer aqah, we added Fig. 17 which shows the average probability of predicting a specific token, clearly illustrating how the GIT-STORM head presents higher probabilities and therefore is more confident in predicting the following tokens. This result corroborates, even more, the idea that the MaskGIT head works well and that in practice there is not a gap between using a masked version of $z_t$ and of $z_{t+1}$.
> > > > > > >
> > > > > > > SQ2: If the above is true, then in Line 3&4 of Algo 2 (called from Line 6 of Algo 1 with $\gamma =1$ ), is the input of the MaskGIT's BidirectionalTransformer a masked version of $z^0$, namely a masked version of $z_{t+1}?
> > > > > > >
> > > > > > > A2: Using the same explanation from A3, the intermediate representations $z^{\gamma}$ can be considered as an in-distribution regime and therefore interpreted as a masked version of $z_t$.
> > > > > > >
> > > > > > >
> > > > > > > [1] W. Yan et al., Temporally Consistent Transformers for Video Generation, 2022
> > > > > > > [2] D. Lee et al., Draft-and-Revise: Effective Image Generation with Contextual RQ-Transformer, 2022

---

> > > > > > > > ### Author Response · Authors · 2024-12-01
> > > > > > > >
> > > > > > > > Dear Reviewer,
> > > > > > > >
> > > > > > > > We did our best to answer all your questions, and I must say we really enjoyed it. Thanks to you we double checked every thought or component of our model. Moreover you introduced as to the concept of weight tying.
> > > > > > > >
> > > > > > > > There is not much time before the end of the rebuttal period, considering you are the only one with a grade of 3, we really hope you increase the grade. At this point if this paper will be accepted or not is in your hands, so we really hope we convinced you about the soundness and the contribution that this paper brings to the community. We did our best and used a lot of compute in this paper to perform experiments on both Atari and DMC. we really hope to see this paper published at ICLR2025.
> > > > > > > >
> > > > > > > > Thank you so much again for all your feedback and time you spent for this review.

---

> ### Author Response · Authors · 2024-11-25
> **Q&A 3**
>
> Q3: Do you mean that due to improved computational efficiency, you can set a larger update-to-data ratio (related to Update agent every env step in Table 8) thus improve the sample efficiency of model-based RL? Can you provide this hyperparameter of GIT-STORM and baselines for a direct comparison? For example, is the sampling efficiency improved because GIT-STORM has a larger "Update agent every env step"? If not, then the KV cache does not seem to contribute to sampling efficiency.
>
> A3.0: We thank the reviewer for asking clarification about the previous answer. We would like to answer the question starting from an analysis of computational complexity and FLOPs computation in transformers with and without KV caching. For the sake of this discussion we only focus on the Autoregressive Transformer, the module where KV caching is used.
>
>
> Here, https://r4j4n.github.io/blogs/posts/kv/ , we can find a FLOPs comparison of vanilla Transformer and Transformer with KV Cache, which explicitly compares the FLOPs spent by vanilla Transformers versus the one by KV Cached Transformers.
>
> By following the article, it can be seen that we can express the two FLOPs as:
>
> Vanilla Transformer $FLOPS = n×(24bsh^2 +4bs^2h)+2bshV$
>
> KV Cache Transformer  $FLOPS = n×(24bh^2 +4bh + 4b KV_{length})+2bhV$
>
> where,
>
> b = Batch Size = 256; s = Sequence Length = 16; h = Hidden Dimension = 128; n = number of layers = 2; V = vocabulary dimension = 32; KV_length = Sequence Length = 16
>
> By substituting our hyperparameters into the equations we can see that:
>
> Vanilla Transformer FLOPs =  3.254.779.904 FLOPs > KV Cache Transformer FLOPs = 2.037.186.656 FLOPs
>
> And that:
>
> KV Cache Transformer FLOPs = 0,625 Vanilla Transformer FLOPs.
>
>
> From these results, we can see that the autoregressive transformer with KV cache is certainly more efficient than the vanilla one, which does not use KV caching.
>
> Equipped with these new results, we now answer explicitly your questions:
> Q3.1: Do you mean that due to improved computational efficiency, you can set a larger update-to-data ratio (related to Update agent every env step in Table 8) thus improve the sample efficiency of model-based RL?
>
> A3.1: The Update Agent Every env step hyperparameter measures how many env steps need to be executed before updating the agent. Sample efficiency is a measure related to the number of imagined trajectories that the model can generate at a given time. Therefore, Sample Efficiency and the Update Agent Every env step hyperparameter are not directly related, in the sense that modifying the hyperparameter will not affect the sample efficiency of the model. However, you are correct when saying that due to improved computational efficiency, we can set a larger update-to-data ratio and improve the sample efficiency of the model. Where with a larger update-to-data ratio we mean that less time is required to generate (or imagine) the data (batch of imagined trajectories) required to make an update of the agent.
>
>
> Q3.2: Can you provide this hyperparameter of GIT-STORM and baselines for a direct comparison?
>
> A3.2: Besides STORM for which the same value is used, the mentioned hyperparameter is not defined consistently across the other different baselines. For instance, DreamerV3 uses a hyperparameter called train ratio, which is used as follows:
>
> args.batch_size = 16
> args.batch_length = 65
> args.replay_context = 1
> args.train_ratio = 32
>
> batch_steps = args.batch_size * (args.batch_length - args.replay_context) = 1/4
> should_train = embodied.when.Ratio(args.train_ratio / batch_steps)
>
> which means that the model is updated every 4 interactions with the environment.
>
> Since studying the effect of this hyperparameter would have not added any evidence supporting our claims, we did not do an ablation study or grid search of this hyperparameter and instead, we consistently used the same value of STORM (Update agent every env step =1).
>
>
> Q3.3: For example, is the sampling efficiency improved because GIT-STORM has a larger "Update agent every env step"?If not, then the KV cache does not seem to contribute to sampling efficiency.
>
> A3.3:
>
> We clarified that the Update agent every env step hyperparameter is not related to the sample efficiency of the method. However, as shown in A3.0, using the KV caching increases the computational efficiency of the autoregressive transformer that is used to generate the imagined trajectories, and therefore, the more efficient the autoregressive transformer, the quicker the generation (or sampling) of imagined trajectories.
>
>
>
> We hope these answers address your questions. Should you have any remaining doubts, we are happy to answer!

---

> ### Comment · Reviewer_JuLK · 2024-12-01
>
> I appreciate the patient response from the authors. I have tried my best to understand, but I am still not fully convinced by the soundness of the MaskGIT head design. Given the overall quality of this paper, I decided to update my score to 5 and confidence to 4.
>
> I remain open to exchanging ideas with AC and other reviewers in the next stage of the reviewing process.

---

> > ### Author Response · Authors · 2024-12-01
> >
> > Dear Reviewer,
> >
> > We really appreciate you changed the grade. Please tell us if there is anything or any other point we can expand, to make the explanation more sounding. It would really help a lot to have a 6. If there is anything we can do it to get there, we’ll do it.
> >
> > Thank you so much for your feedback and time! And thank you so much for raising the score!

---

### Official Review · Reviewer_aqah · 2024-10-27

**Soundness:** 3
**Presentation:** 3
**Contribution:** 2
**Rating:** 6
**Confidence:** 4

**Summary:**

This paper proposed to replace the MLP head with MaskGIT prior in STORM to achieve a higher quality of latent generation, and therefore achieve better performance on the Atari100k benchmark.
This paper also bridges the gap of the lack of evaluation of transformer-based world models on continuous control tasks.

**Strengths:**

1. This paper clearly distinguishes itself from previous work, with good comparison and illustration.

2. One-hot categorical latent is widely used in recent model-based RL, yet the research on it is insufficient. This paper provides a novel view of it.

3. This paper bridges the gap of the lack of evaluation of transformer-based world models on continuous control tasks.

**Weaknesses:**

1. The motivation and effect of using MaskGIT head in world models are unclear.
Is there any evidence that the world models would have hallucinations, and how could a MaskGIT head mitigate such issues?
How to distinguish if the improved performance (both RL and FVD) comes from more parameters or the MaskGIT prior?

    There should be some further investigation into the mechanism of the MaskGIT head. Such as:

    (a) What's the difference between the latent variables (or distributions) generated with the MLP head and MaskGIT head?

    (b) This MaskGIT head looks like a skip/residual connection from $z_{t}$ to $\hat{z}_{t+1}$, would this reduce the KL divergence in the training or imagination?

    (c) These are sample questions. An investigation like this would improve the soundness and contribution of this paper.

2. Section 2.1 could be more concise, as these are not quite related to the key contributions and are frequently repeated in each of the model-based RL papers.

**Questions:**

1. On lines 307-309, I think STORM uses KV caching in both the conditioning phase and the imagination phase, see [here](https://github.com/weipu-zhang/STORM/blob/e0b3fd44320d7e213ec905c673ad3f35b61b89f4/sub_models/world_models.py#L363). The `predict_next()` uses `forward_with_kv_cache()` for decoding.

2. Missing comma on line 214?

3. What's new in the proposed state mixer compared to the STORM's action mixer?

4. `Freeway` is a hard exploration environment, as the agent has to repeat the `up` operation many times to get a first reward, which is a rare event for a random policy. Without the first reward, the value space is all zero and the policy would be further optimized as a uniform random policy. STORM, IRIS, and DIAMOND have different tricks that can mitigate such an issue. But what is the underlying reason for GIT-STORM to reach a non-zero result? I think this is not related to the improved decoding or world modelling quality since DreamerV3 and STORM (w/o traj) could also produce a nearly perfect reconstruction and prediction on `Freeway`.

5. For the `Quadruped Run` in Figure 6, I wonder if it's too small (compared to Figure 4 in [DreamerV3](https://arxiv.org/pdf/2301.04104)).

6. Lines 529-530, "Replacing...", the order is reversed.

---

> ### Author Response · Authors · 2024-11-23
> **Q&A 1**
>
> We thank the reviewer for such insightful and important questions. We are really happy to answer and fulfil any remaining doubts. Here we list aggregated questions and the related answers.
>
> Q1: The motivation and effect of using MaskGIT head in world models are unclear. Is there any evidence that the world models would have hallucinations, and how could a MaskGIT head mitigate such issues? How to distinguish if the improved performance (both RL and FVD) comes from more parameters or the MaskGIT prior?
>
> A1: We thank the reviewer for the relevant question. Hallucinations are associated with any autoregressive model that uses sampling to compute the next token, such as GPT models, but also transformer-based world models. When it comes to world models, hallucinations may appear in the form of unrealistic reconstructions or wrong state dynamics transitions.  Table 1 shows the comparison of TECO between using an MLP prior and
> a MaskGIT prior for video prediction in terms of FVD, where it can be seen that using a MaskGIT prior leads to much better video prediction performances (TECO w/ MaskGIT: 48 -> TECO w/ MLP head: 153 and TECO w/ MaskGIT: 199 -> TECO w/ MLP head: 228; respectively on DMLab and SSv2). This finding motivated us to explore the use of a MaskGIT head in world models. Indeed, since the MaskGIT leads to better results in Video Prediction tasks, it is plausible that it increases the ability of the dynamics module to produce better state transitions, which are the main source of hallucinations in world models and video prediction models.
>
> The MaskGIT head has actually way fewer parameters than an MLP one ( MLP head: 4720640 parameters  -> MaskGIT head: 460960 parameters). Therefore, it cannot be the case that the number of parameters is responsible for the improvement in performance. By contrast, the difference in number of parameters shows that the MaskGIT head presents a better inductive bias and requires fewer parameters to perform better.

---

> ### Author Response · Authors · 2024-11-23
> **Q&A 2**
>
> Q2: There should be some further investigation into the mechanism of the MaskGIT head. Such as: (a) What's the difference between the latent variables (or distributions) generated with the MLP head and MaskGIT head? (b) This MaskGIT head looks like a skip/residual connection from to, would this reduce the KL divergence in the training or imagination? (c) These are sample questions. An investigation like this would improve the soundness and contribution of this paper.
>
> A2: We greatly appreciate the reviewer’s insightful suggestion regarding further investigation into the mechanism of the MaskGIT head. Below, we address the proposed questions and provide additional analysis to strengthen the paper’s contribution:
>
> (a) Difference between latent variables (or distributions) generated by the MLP head and the MaskGIT head
>
> The MaskGIT head fundamentally differs from the MLP head in how it generates token distributions. Figure 17 provides a visualization of the logits distributions produced by the dynamics head in both GIT-STORM and STORM. A closer inspection reveals that both MLP and MaskGIT heads generate mean distributions with two distinct peaks: one near zero, corresponding to tokens that are unlikely to be sampled, and another, smaller peak, which captures the confidence of sampling a given token. This dual-peak structure reflects the model's ability to differentiate between likely and unlikely tokens, offering finer granularity in token sampling decisions. Therefore, the higher the second peak and the broader the distribution's support, the more confident the world model is in sampling tokens for a given dynamics state transition. Consistent with the perplexity values presented in Table 3, GIT-STORM produces more refined probability distributions, enabling it to make predictions with greater confidence compared to STORM.
>
> (b) Relationship between the MaskGIT head and KL divergence
>
> As illustrated in Figure 16, GIT-STORM demonstrates consistently lower KL divergence across three environments (Hero, Boxing, and Freeway) compared to STORM. This reduction in KL divergence can be attributed to the MaskGIT head, which resembles a residual connection by leveraging token embeddings to compute logits. This approach ensures that token distributions are closer to the true dynamics of the environment, leading to more efficient encoding of state transitions. The residual-like design minimizes redundant learning, resulting in reduced KL divergence.
>
> Notably, the most significant improvement is observed in the Boxing environment, where GIT-STORM exhibits a marked advantage. This finding aligns with the hypothesis that the MaskGIT head facilitates better representation learning for complex dynamics.
>
> (c) Contribution of this investigation to the paper’s soundness
>
> To further elucidate the contribution of the MaskGIT head, we have incorporated both KL divergence comparisons (Figure 16) and visualizations of the dynamics head’s output distributions (Figure 17). These analyses collectively provide insights into how the MaskGIT head improves the dynamics state transition modeling.
>
> In summary, the MaskGIT head not only enhances the accuracy of token distributions but also promotes better sequence modeling by reducing KL divergence. This investigation underscores its critical role in achieving the improved performance of GIT-STORM and enriches the paper’s contribution by offering a deeper understanding of its design choices.

---

> ### Author Response · Authors · 2024-11-23
> **Q&A 3 and 4**
>
> Q3: Section 2.1 could be more concise, as these are not quite related to the key contributions and are frequently repeated in each of the model-based RL papers.
>
> A3: We thank the reviewer for this feedback. We agree that the section could be a bit more concise. We removed some of the information that can be easily found in any other paper within the field and that does not directly relate to our work.
>
> Q4: On lines 307-309, I think STORM uses KV caching in both the conditioning phase and the imagination phase, see here. The predict_next() uses forward_with_kv_cache() for decoding.
>
> A4: We thank the reviewer for noticing this mistake. Indeed STORM uses KV caching already. What we did was vectoring the for loop within the conditioning phase, since all information is available from the start. One of the authors misstated the contribution. We removed the lines that relate to KV caching contributions in our paper. We apologize for the mistake.

---

> ### Author Response · Authors · 2024-11-23
> **Q&A 5 and 6**
>
> Q5: Freeway is a hard exploration environment, as the agent has to repeat the up operation many times to get a first reward, which is a rare event for a random policy. Without the first reward, the value space is all zero and the policy would be further optimized as a uniform random policy. STORM, IRIS, and DIAMOND have different tricks that can mitigate such an issue. But what is the underlying reason for GIT-STORM to reach a non-zero result? I think this is not related to the improved decoding or world modelling quality since DreamerV3 and STORM (w/o traj) could also produce a nearly perfect reconstruction and prediction on Freeway.
>
>
> A5: We thank the reviewer for such an insightful question. Although it is true that exploration strategy plays a big role in this case, within the scope of this paper we did not consider this aspect and focused instead on the representation learning perspective of the world model. Indeed, in STORM [2], Fig.4 shows that using different policy input spaces significantly affects the learning behavior.
>
> This result suggests that besides the exploration strategy, the policy input space plays a role when it comes to policy learning behavior. Therefore, we believe that the GIT-STORM policy input space allows the policy to learn effectively a meaningful policy in Freeway.
>
> Q6: For the Quadruped Run in Figure 6, I wonder if it's too small (compared to Figure 4 in DreamerV3).
>
> DreamerV3 upsampled the output images for visualization purposes, unfortunately, due to time constraints we could not do the same. We will try to update the image before the end of the review period, for now we removed the sequence.

---

> ### Comment · Reviewer_aqah · 2024-11-23
>
> The authors have addressed most of my concerns. Given the limited research on latent space in MBRL and the novel structure of GIT-STORM, I believe this paper should be accepted, and I will increase my score to 6.
>
> However, since the paper primarily offers an incremental contribution and does not deeply investigate the latent space to provide design guidance for future world models, I cannot justify a score of 8.
>
> Additionally, regarding the Quadruped Run task, the camera configuration could perhaps reference [this implementation](https://github.com/NM512/dreamerv3-torch/blob/main/envs/dmc.py). That said, it’s not particularly important at this stage.

---

> > ### Author Response · Authors · 2024-12-01
> >
> > We really thank the reviewer for your feedback and suggestions!
> >
> > Do you have in mind any particular analysis we should carry out with respect to the latent space?

---

### Official Review · Reviewer_Lr34 · 2024-10-30

**Soundness:** 2
**Presentation:** 2
**Contribution:** 2
**Rating:** 5
**Confidence:** 4

**Summary:**

The paper introduces GIT-STORM, which incorporates three modifications to the base algorithm STORM: a MaskGIT prior that replaces the MLP dynamics head, a draft-and-revise decoding scheme for enhanced consistency, and a state mixer for continuous action environments. Experimental results demonstrate that GIT-STORM surpasses STORM on both Atari 100K and DMC benchmarks. Video prediction results indicate that this improvement is attributed to more accurate representations.

**Strengths:**

- The motivation of incorporating a MaskGIT prior into the STORM architecture is clear.
- The proposed method is straightforward and easy to reproduce.
- MaskGIT can effectively improve the video prediction quality of STORM,  indicating applicability of GIT-STORM to more complicated tasks.

**Weaknesses:**

- The paper contains a misstatement in its contributions. The authors claim that they "apply transformer-based world models to continuous action environments for the first time". This claim is inaccurate, as TransDreamer[1] can also be applied to continuous action environments. The authors are evidently aware of this paper, given that they have cited it in this work.
- The state-mixer design is not properly addressed. If the authors claim this part of their contribution, they should either elaborate on the design, or provide empirical results to show the superiority of this method. Based on the overlapping tasks, TransDreamer appears to have better performance than GIT-STORM+state-mixer on the continuous control benchmark DMC.
- The experimental results in Atari 100K only demonstrate marginal improvement. The gain over STORM seems to primarily originate from the gopher task alone, which contains inconsistent results, as detailed in the questions section.

[1] Chen et al. TransDreamer: Reinforcement Learning with Transformer World Models.

**Questions:**

- Results on the Freeway task have very high variance according to Figure 10. How many out of the five runs does GIT-STORM actually achieve non-zero performance?
- The most challenging aspect of learning the Freeway task is obtaining the first successful trajectory, which I believe is more related to the exploration strategy than state predictions, given the sparse rewards. How does GIT-STORM assist the agent in exploring more efficiently? Is this strategy stable, or are the successful trajectories obtained by random seeds?
- Why would the pendulum swingup task fail for both STORM and GIT-STORM? DreamerV2, DreamerV3 and TransDreamer can learn this task fairly easily.
- The experiment results in Table 5 and Figure 10 appear inconsistent. For instance, the Gopher score reported in Table 5 is 8562, but the last point in Figure 10 shows a performance of around 2500. Do these two results use different metrics?
- Could you add the learning curves of STORM or DreamerV3 to Figure 10 for a better comparison, considering that you have reproduced these results?

---

> ### Author Response · Authors · 2024-11-23
> **Q&A 1**
>
> We thank the reviewer for such insightful and important questions. We are really happy to answer and fulfil any remaining doubts. Here we list aggregated questions and the related answers.
>
> Q1: The paper contains a misstatement in its contributions. The authors claim that they "apply transformer-based world models to continuous action environments for the first time". This claim is inaccurate, as TransDreamer[1] can also be applied to continuous action environments. The authors are evidently aware of this paper, given that they have cited it in this work.
>
> A1: We thank the reviewer for this insightful feedback. We indeed agree that our statement is too big of a claim considering that TransDreamer [1] shows some results on a subset of DMC environments. However, there are also a couple of reasons that led us to use this statement:
>
> 1) TransDreamer was never published at a conference, the submission was withdrawn and the authors stated:
>
> “Since submission, we have found a few aspects of the paper we want to improve including some points mentioned by the reviewers. We have decided to withdraw our paper and improve it for a future conference. Thank you to the reviewers who spent the time reading our paper and providing useful feedback.”
>
> 2) The source code was never released, which made it difficult to reproduce the results
>
> 3) There is not a detailed explanation of how latent representations z and actions are combined. Moreover, it is also not clear which encoder was used to extract the representations z. However, by looking at the TransDreamer Loss function, it seems that the latent representations are continuous.
>
> These three points make clear that, although TransDreamer is a transformer-based architecture, it does not use discrete representations (or tokens).
> Usually, discrete representations (e.g., tokens) are the main reason why it is hard to work with continuous actions when working with transformer-based world models. Moreover, while continuous latent representations may enable an easier integration with continuous actions, they also decrease substantially the sample efficiency of the proposed approach. As a result, they were not able to validate and benchmark the model on many different environments and, instead, they focused only on a subset of DMC and Atari.
> Considering also that the premise of using transformer-based architecture is that they will enable a better sample efficiency [2], we believe that since the TransDreamer approach does not use discrete representations, it is out of scope concerning our discussion and proposed approach. However, thanks to your feedback, we now understand that this may be misleading and is not exactly correct. Therefore, we will change the statement from:
>
> "apply transformer-based world models to continuous action environments for the first time”.
>
> To:
>
> "apply categorical transformer-based world models to continuous action environments for the first time”.
>
> Highlighting that we are solving the problem related to this subset of models, which however is also the most relevant in our community. Indeed, iVideoGPT [3], a transformer-based discrete world model, was the first-ever world model trained on the Open X-Embodiment dataset [4], an extremely big robotics dataset built from 22 different robots collected through a collaboration between 21 institutions, demonstrating 527 skills (160266 tasks).
> Moreover, iVideoGPT authors state:
> “iVideoGPT is scalable for action-free video prediction across a mixture of over one million robotic and human manipulation trajectories.”
> This sentence, together with the whole introduction and methodology sections, shows how using a scalable model (e.g., transformers using discrete tokens) to learn a world model allows us to tackle way harder and more realistic environments than game environments. Such scalability has been achieved only using categorical transformer-based world models, which is why we decide to focus on this class of models.
>
> Finally TransDreamer reviewer AUKA stated:
> “the paper just reports 'it works' but lacks meaningful insights or observations that come from utilizing Transformers instead of RNNs.”, https://openreview.net/forum?id=s3K0arSRl4d
>
> We would like to add that our goal is to inform the community with meaningful insights and observations about the usage of masked generative priors and continuous action integration in discrete transformer-based world models. Indeed, in this paper with try to explicitly explain the effect of these components by showing their effect also on metrics like FVD and perplexity, providing different angles to evaluate how these components improve the sequence modelling capabilities of transformer-based world models.
>
>
>
>
> [2] Weipu Zhang et al. STORM: Efficient Stochastic Transformer based World Models for Reinforcement Learning, 2024. https://arxiv.org/abs/2310.09615

---

> ### Author Response · Authors · 2024-11-23
> **Q&A 2**
>
> Q2: The state-mixer design is not properly addressed. If the authors claim this part of their contribution, they should either elaborate on the design, or provide empirical results to show the superiority of this method. Based on the overlapping tasks, TransDreamer appears to have better performance than GIT-STORM+state-mixer on the continuous control benchmark DMC.
>
>
> A2: We thank the reviewer for the feedback. Although we do not use a different state mixer design, as you suggest, in Appendix E we show an ablation comparing three different state mixer designs. Particularly, we compare the original design (concatenation of z and a) against adding an attention mechanism after concatenation or just computing cross attention between z and a, to evaluate different inductive biases to design an optimal state mixer. Figure 13, shows that the original design is the one that works better, which is why we sticked to it.
>
> Although we left as future work a more extensive exploration of state mixer designs, this experiment let us conclude that using a very simple inductive bias is the most effective solution because when working with world models latent spaces, by using a more elaborate inductive bias, we may introduce an implicit constraint that ultimately deteriorates the representation space and, therefore, the downstream results (e.g., scores).
>
> Moreover, we added an ablation of the State Mixer to assess how the action mixer contributes to the presented performances, comparing a setting with unmixed states (which uses concatenated states and actions but does not use a mixer) and iVideoGPT action embedding approach.
> Figure 14 demonstrates that the State Mixer consistently outperforms the considered baselines. Interestingly, under the given setup, the iVideoGPT approach fails to learn meaningful policies. We hypothesize that this limitation arises from the scale of the training procedure and considered environments. Specifically, iVideoGPT is designed to leverage much larger datasets, enabling it to learn robust representations.
>
> Moreover, we observe that bypassing the State Mixer by directly concatenating and feeding state and action embeddings into the transformer allows the model to learn policies that are meaningful but perform suboptimally compared to the State Mixer-based approach. This finding highlights the effectiveness of the State Mixer in extracting and processing state-action representations crucial for learning optimal policies.
>
> Finally, for the same reasons explained in A1, we do not include TransDreamer in the benchmark.
>
> We are more than happy to answer and address any other questions if you have any remaining doubts or concerns.

---

> ### Author Response · Authors · 2024-11-23
> **Q&A 3**
>
> Q3: The experimental results in Atari 100K only demonstrate marginal improvement. The gain over STORM seems to primarily originate from the gopher task alone, which contains inconsistent results, as detailed in the questions section.
>
>
> A3: We understand the confusion that the results table may bring, and we hope to address any doubts you may have regarding this question here.
> In Deep Reinforcement Learning at the Edge of the Statistical Precipice: https://proceedings.neurips.cc/paper/2021/hash/f514cec81cb148559cf475e7426eed5e-Abstract.html , the authors write:
> “Only reporting point estimates obscures nuances in comparisons [85] and can erroneously lead the field to conclude which methods are state-of-the-art [63, 84], ensuing wasted effort when applied in practice [108].”
>
> Therefore, the results and comparisons within single environments cannot be considered to define which model is state-of-the-art. Instead, Optimality GAP, Interquantile Mean and Probability of improvement are statistically significant metrics and are the ones to be considered. We realized there was a bug in the plotting function we used and some seeds were not considered. We fixed it and now the inconsistencies are not there anymore.
>
>
>
> [63] Jimmy Lin, Daniel Campos, Nick Craswell, Bhaskar Mitra, and Emine Yilmaz. Significant improvements over the state of the art? a case study of the ms marco document ranking leaderboard. arXiv preprint arXiv:2102.12887, 2021.
> [84] Nils Reimers and Iryna Gurevych. Reporting score distributions makes a difference: Performance study of lstm-networks for sequence tagging. In Proceedings of the 2017 Conference on Empirical Methods in Natural Language Processing, pages 338–348, 2017.
> [85] Samuel Ritter, David GT Barrett, Adam Santoro, and Matt M Botvinick. Cognitive psychology for deed neural networks: A shape bias case study. In International conference on machine learning, 2017.
> [108] Gaël Varoquaux and Veronika Cheplygina. How i failed machine learning in medical imaging–shortcomings and recommendations. arXiv preprint arXiv:2103.10292, 2021.

---

> ### Author Response · Authors · 2024-11-23
> **Q&A 4 and 5**
>
> Q4: Results on the Freeway task have very high variance according to Figure 10. How many out of the five runs does GIT-STORM actually achieve non-zero performance?
>
> A4: We thank the reviewer for this question. It is true that the results presents a high-variance, this happens because 3 out of 5 seeds presents non-zero results.
>
> Q5: The most challenging aspect of learning the Freeway task is obtaining the first successful trajectory, which I believe is more related to the exploration strategy than state predictions, given the sparse rewards. How does GIT-STORM assist the agent in exploring more efficiently? Is this strategy stable, or are the successful trajectories obtained by random seeds?
>
> A5: We thank the reviewer for such an insightful question. Although it is true that exploration strategy plays a big role in this case, within the scope of this paper we did not consider this aspect and focused instead on the representation learning perspective of the world model. Indeed, in STORM [2], Fig.4 shows that using different policy input spaces affects the learning behavior.
>
> This result suggests that besides the learning strategy, the policy input space plays a role when it comes to policy learning behavior. Therefore, we can say that GIT-STORM policy input space allows the policy to learn effectively a meaningful policy.

---

> ### Author Response · Authors · 2024-11-23
> **Q&A 6,7 and 8**
>
> Q6: Why would the pendulum swingup task fail for both STORM and GIT-STORM? DreamerV2, DreamerV3 and TransDreamer can learn this task fairly easily.
>
> A6: As described in A1, Transdreamer and DreamerV3 use continuous representations (together with stochastic discrete counterpart), to represent frames. We believe this to be the main difference and reason why they can learn this task fairly easily, while GIT-STORM can’t. However, we would like to remind the reviewer of all the advantages of using discrete representations instead as described in A1.
>
> Q7: The experiment results in Table 5 and Figure 10 appear inconsistent. For instance, the Gopher score reported in Table 5 is 8562, but the last point in Figure 10 shows a performance of around 2500. Do these two results use different metrics?
>
> A7: We thank the reviewer for such detail-oriented feedback. While Table 5 shows the evaluation results, Figure 10 shows the training profiles. Moreover, the profiles were plotted using 3 different seeds, while Table 5 was made using 5 different seeds. We adjusted the inconsistency and updated the plot, which now is much more coherent.
>
> Q8: Could you add the learning curves of STORM or DreamerV3 to Figure 10 for a better comparison, considering that you have reproduced these results?
>
> A8: We thank the reviewer for the feedback. We added the STORM learning curve.

---

> > ### Author Response · Authors · 2024-12-01
> > **Reminder**
> >
> > Dear Reviewer,
> >
> > Could you please answer us? We addressed every question you had and weaknesses you mentioned.
> >
> > We only have 2 days before the time to discuss ends. We kindly ask you to reply, we would really appreciate to have the chance to address any remaining concerns or doubts. On the other hand, if you don’t have any, we would be really happy if you could raise our grade, considering we addressed to all your questions.
> >
> > We really put a lot of work on this project, we really hope we can hear from you before the time ends.

---

### Official Review · Reviewer_zpZw · 2024-11-01

**Soundness:** 2
**Presentation:** 3
**Contribution:** 2
**Rating:** 6
**Confidence:** 4

**Summary:**

The paper, Masked Generative Priors Improve World Models Sequence Modelling Capabilities, introduces GIT-STORM, an extension of the STORM architecture, incorporating MaskGIT as a dynamics prior to enhance sequence modeling in world models. The authors address two main gaps in previous research: the limitation of transformer-based world models in continuous action environments and the inadequacies of prior methods, like STORM, in capturing effective state representations. Through experiments on Atari 100k (discrete) and DeepMind Control Suite (continuous), GIT-STORM demonstrates improvements in RL and video prediction, suggesting that Masked Generative Priors could be a powerful inductive bias for world models, supporting broader applicability across diverse RL tasks and environments.

**Strengths:**

The empirical evaluation spans discrete and continuous action benchmarks, providing a robust assessment of GIT-STORM’s performance. The reported results demonstrate that GIT-STORM not only improves sample efficiency in RL tasks but also enhances video prediction quality, particularly in the Atari 100k benchmark, aligning well with the study's objectives. Moreover, the paper is well-written with a clear structure, providing a good experience as a reader. Extending the transformer-based world models to continuous action tasks also poses a sufficient novelty and broadens the utility of these models in RL and video prediction applications.

**Weaknesses:**

It remains unclear why GIT-STORM does not consistently outperform STORM across all benchmarks or why it fails to close the performance gap with DreamerV3 in environments beyond Atari 100k. The paper does not fully explain the conditions under which GIT-STORM’s improvements are more marginal, suggesting a need for clearer insights into the impact of individual architectural components.

The paper claims state-of-the-art results for GIT-STORM on select environments, yet Table 6 seems to indicate that DrQ-v2 outperforms GIT-STORM on two environments (where the authors claim they are better?). Clarifying the conditions under which GIT-STORM achieves these results or adjusting the claim would help ensure consistency and accuracy in presenting the model's achievements.

The proposed approach for handling continuous action spaces is promising, yet lacks a comprehensive empirical analysis. Additional studies on more diverse continuous control tasks could provide stronger validation of the state mixer function's effectiveness and the broader applicability of the model in continuous settings. Most importantly, the modifications from STORM to GIT-STORM are extensive, involving MaskGIT, state mixer, policy adjustments from DreamerV3, and an observation module from STORM. The compounded modifications make it difficult to discern the exact contribution of each component to the reported performance improvements. A more focused ablation study could is required to isolate the impact of each modification.

**Questions:**

- Could the authors elaborate on why GIT-STORM occasionally does not surpass STORM and the conditions where improvements are only minor? Understanding this would clarify the contextual efficacy of the MaskGIT prior.
- Regarding the reported state-of-the-art claim, Table 6 suggests that DrQ-v2 outperforms GIT-STORM in some highlighted environments. Could the authors comment why they claim GIT-STORM provides SOTA results on these? It is not the case, right?
- What is the rationale for improving STORM over directly utilizing DreamerV3, which appears to perform better in many scenarios? Or put differently: why would one care to improve STORM with the proposed modifications when there is DreamerV3 and I could just use it or improve over DreamerV3?

I am open to increase my score once there is clarity on these questions.

---

> ### Author Response · Authors · 2024-11-14
> **Q&A 1**
>
> We thank the reviewer for such insightful and important questions. We are really happy to answer and fulfill any remaining doubts. Here we list aggregated questions and the related answers.
>
>
> Q1: The paper does not fully explain the conditions under which GIT-STORM’s improvements are more marginal, suggesting a need for clearer insights into the impact of individual architectural components. Most importantly, the modifications from STORM to GIT-STORM are extensive, involving MaskGIT, state mixer, policy adjustments from DreamerV3, and an observation module from STORM. The compounded modifications make it difficult to discern the exact contribution of each component to the reported performance improvements. A more focused ablation study could be required to isolate the impact of each modification.
>
>
>
> A1: We agree with the reviewer; we are working on an ablation study that assesses the contribution of the following components:
>
> Maskgit head: We compare the MaskGIT head to the MLP head.
> Logits computation using the dot product between $\xi_t$ and MaskGit embeddings: We compare the GIT-STORM setup against using the MLP head that takes as input eps_t and returns logits, validating how using the dot product between eps_t and MaskGit embeddings impacts the performances.
> Mixed states: we compare using a state mixer against using the unmixed states as world model autoregressive transformer input. As a result, we can validate to what extent the capacity introduced by the state mixer affects GIT-STORM performances.
>
>
> Following your suggestions, we believe that these are the key components to analyze to understand GIT-STORM capabilities and the exact contribution of each component to the reported performance improvements.
>
>
>
> We will validate the ablation on 3 Atari games (Hero, Freeway, and Boxing) and 3 DMC environments (Walker Walk, Walker Run, and Quadruped Run) over 3 seeds. We will send the ablations as soon as possible.

---

> ### Author Response · Authors · 2024-11-14
> **Q&A 2**
>
> Q2: It remains unclear why GIT-STORM does not consistently outperform STORM across all benchmarks. Could the authors elaborate on why GIT-STORM occasionally does not surpass STORM and the conditions where improvements are only minor? Understanding this would clarify the contextual efficacy of the MaskGIT prior.
>
>
>
> A2: We understand why this may be confusing and thank the reviewer for asking this question.
>
> In Deep Reinforcement Learning at the Edge of the Statistical Precipice: https://proceedings.neurips.cc/paper/2021/hash/f514cec81cb148559cf475e7426eed5e-Abstract.html , the authors write:
>
> “Only reporting point estimates obscures nuances in comparisons [85] and can erroneously lead the field to conclude which methods are state-of-the-art [63, 84], ensuing wasted effort when applied in practice [108].”
>
> Therefore, the single values of each run cannot be really considered singularly, as the variation and stochasticity of these runs depend on a multitude of factors (e.g., used random seed, luck in exploration, etc.). Hence, whether GIT-STORM consistently outperforms STORM can only be assessed from a statistical perspective. In particular, following “Deep Reinforcement Learning at the Edge of the Statistical Precipice”, we use the metrics they propose to meaningfully assess models' capabilities in this field (e.g., mean, median, IQM, optimal GAP, and probability of improvement over other baselines P(GIT-STORM>Y), where Y are all other baselines). A more detailed explanation of these metrics can be found in Appendix J.
>
> According to these metrics, GIT-STORM consistently outperforms STORM.
>
> Finally, not only the single values are not reliable but also assessing why a model performs better than another one cannot be done with confidence or rational arguments. That’s why rather than writing conjectures on why GIT-STORM outperforms STORM in single environments, we base our arguments on the statistical metrics.
>
> We hope this addresses your question, but we are more than happy to keep discussing should there be any other concerns or doubts.

---

> ### Author Response · Authors · 2024-11-14
> **Q&A 3**
>
> Q3: The paper claims state-of-the-art results for GIT-STORM on select environments, yet Table 6 seems to indicate that DrQ-v2 outperforms GIT-STORM on two environments (where the authors claim they are better?). Regarding the reported state-of-the-art claim, Table 6 suggests that DrQ-v2 outperforms GIT-STORM in some highlighted environments. Could the authors comment on why they claim GIT-STORM provides SOTA results on these? It is not the case, right?
>
>
>
> A3: We understand the confusion that the results table may bring, and we hope to address any doubts you may have regarding this question here.
>
> In Deep Reinforcement Learning at the Edge of the Statistical Precipice: https://proceedings.neurips.cc/paper/2021/hash/f514cec81cb148559cf475e7426eed5e-Abstract.html , the authors write:
>
> “Only reporting point estimates obscures nuances in comparisons [85] and can erroneously lead the field to conclude which methods are state-of-the-art [63, 84], ensuing wasted effort when applied in practice [108].”
>
> Moreover, also because of these reasons, within this field the results within the 5% of variation of state-of-the-art (SOTA) values are also considered SOTA performances [1].
>
>
>
> Therefore, the results and comparisons within single environments cannot be considered to define which model is state-of-the-art, and following the community standards, we defined as state-of-the-art the results where GIT-STORM performs above every other model, or within the 5% of variation from SOTA values. That’s why we state:
>
> “GIT-STORM presents state-of-the-art scores on two environments, Walker Stand, and Quadruped Run.”
>
>
>
> [1] Danijar Hafner, Jurgis Pasukonis, Jimmy Ba, and Timothy Lillicrap. Mastering diverse domains through world models. arXiv preprint arXiv:2301.04104, 2023.
>
> [63] Jimmy Lin, Daniel Campos, Nick Craswell, Bhaskar Mitra, and Emine Yilmaz. Significant improvements over the state of the art? a case study of the ms marco document ranking leaderboard. arXiv preprint arXiv:2102.12887, 2021.
>
> [84] Nils Reimers and Iryna Gurevych. Reporting score distributions makes a difference: Performance study of lstm-networks for sequence tagging. In Proceedings of the 2017 Conference on Empirical Methods in Natural Language Processing, pages 338–348, 2017.
>
> [85] Samuel Ritter, David GT Barrett, Adam Santoro, and Matt M Botvinick. Cognitive psychology for deed neural networks: A shape bias case study. In International conference on machine learning, 2017.
>
> [108] Gaël Varoquaux and Veronika Cheplygina. How i failed machine learning in medical imaging–shortcomings and recommendations. arXiv preprint arXiv:2103.10292, 2021.

---

> ### Author Response · Authors · 2024-11-14
> **Q&A 4**
>
> Q4: Why does it fail to close the performance gap with DreamerV3 in environments beyond Atari 100k. What is the rationale for improving STORM over directly utilizing DreamerV3, which appears to perform better in many scenarios? Or put differently: why would one care to improve STORM with the proposed modifications when there is DreamerV3 and I could just use it or improve over DreamerV3?
>
>
>
> A4: We thank the reviewer for such a fundamental and important question.
>
> The rationale for improving STORM over directly utilizing DreamerV3 comes from the advantages of transformer-based world models over RSSMs, like the one used in Dreamer-based architectures. Indeed, as stated in the Introduction section:
>
> “As RNNs impede parallelised computing due to its recurrent nature, some works (Micheli et al., 2022; Robine et al., 2023; Zhang et al., 2023) have incorporated autoregressive transformer architectures (Vaswani et al., 2017), which have been shown to be effective across various domains, such as language (Devlin et al., 2019; Radford et al., 2019; Brown et al., 2020; Raffel et al., 2020), images (Dosovitskiy et al., 2021; He et al., 2022; Liu et al., 2023), and offline RL (Janner et al., 2021; Chen et al., 2021). For instance, IRIS (Micheli et al., 2022) utilises discrete autoencoders (Oord et al., 2017) to map raw pixels into a smaller set of image tokens to be used as the input to the world model, achieving super-human performance in ten different environments of the Atari 100k benchmark (Kaiser et al., 2019).”
>
>
>
> Expanding this paragraph, transformer-based world models are very promising considering that dynamic state transitions can be performed in parallel. Indeed, iVideoGPT [2], a transformer-based world model, was the first-ever world model trained on the Open X-Embodiment dataset [3], an extremely big robotics dataset built from 22 different robots collected through a collaboration between 21 institutions, demonstrating 527 skills (160266 tasks).
>
> Moreover, iVideoGPT authors state:
>
> “iVideoGPT is scalable for action-free video prediction across a mixture of over one million robotic and human manipulation trajectories.”
>
> This sentence, together with the whole introduction and methodology sections, shows how using a scalable model (e.g., transformers) to learn a world model allows us to tackle way harder and more realistic environments than game environments.
>
> To sum up and answer your question explicitly, pursuing research within the domain of transformer-based transformers rather than using DreamerV3-based models that rely on RSSMs will eventually allow the community to build scalable models that are able to handle real-world environments—the ultimate goal of this community.
>
> Finally, we would like to add that the goal of the paper is to inform the community about the fact that Masked generative priors (e.g., MaskGIT priors) improve the sequence modelling capabilities of transformer-based world models. This, although tested on simpler environments, is something that definitely is worth exploring in more realistic environments and architectures, such as iVideoGPT, which would be a perfect fit for this method. If we had more time and world model weights were available, we would have validated the usage of MaskGIT prior to it as well.
>
>
>
> [2] Jialong Wu and Shaofeng Yin and Ningya Feng and Xu He and Dong Li and Jianye Hao and Mingsheng Long. iVideoGPT: Interactive VideoGPTs are Scalable World Models, 2024, https://arxiv.org/abs/2405.15223.
>
> [3] Open X-Embodiment Collaboration et al. Open X-Embodiment:Robotic Learning Datasets on RT-X Models, 2024, https://robotics-transformer-x.github.io/.

---

> ### Author Response · Authors · 2024-11-23
> **Ablations results - submission update**
>
> We thank the reviewer again for suggesting us to analyze the ablations of the proposed model. We updated the submission and you can now find the ablations in Section E.
>
> Figure 12 illustrates an ablation study on three Atari games (Hero, Freeway, and Boxing) and three DMC environments (Walker Walk, Walker Run, and Quadruped Run). Across both sets of environments, the removal of the MaskGIT head consistently results in poorer downstream performance (e.g., lower scores). Additionally, leveraging the dot product between $\xi_t$ and MaskGIT embeddings has a substantial impact in environments such as Freeway, Walker Walk, and Quadruped Run. However, its influence appears negligible in other environments like Hero and Walker Run, suggesting that its efficacy may be context-dependent.
>
> Please let us know if you have any remaining doubts or concerns.

---

> > ### Comment · Reviewer_zpZw · 2024-11-23
> >
> > Thank you for providing responses to my raised questions and the additional ablation results. I have increased my rating to 6.

---

### Author Response · Authors · 2024-12-02
**Summary of rebuttal for Area Chairs**

Dear Area Chairs,

We appreciate the reviewers' thoughtful evaluations of our paper and have carefully addressed their concerns. Below, we provide a unified discussion of the main issues raised and our responses, followed by the outcomes for each reviewer.

---

Main Concerns and Our Responses:

- Clarification of Contributions and Claims:
  - Concern: Some reviewers noted that our claim of being the first to apply transformer-based world models to continuous action environments was overstated due to existing work like TransDreamer.
  - Response: We acknowledged this oversight and revised our claim to specify that we are the first to apply categorical transformer-based world models to continuous action environments. We highlighted the distinction between our use of discrete (categorical) representations and prior work's continuous representations, emphasizing the unique challenges and contributions of our approach.

- State Mixer Design and Analysis:
  - Concern: The design and effectiveness of the state mixer were not adequately explained or empirically validated.
  - Response: We expanded the description of the state mixer in the paper, detailing how it integrates continuous actions with discrete latent states. We conducted an ablation study comparing different state mixer designs, demonstrating the effectiveness of our approach through empirical results on both Atari and DeepMind Control (DMC) benchmarks. Moreover, we compared the used inductive bias against other SOTA approaches, such as the one used in iVideoGPT, and showed that the approach outperforms all included baselines.

- Use and Mechanism of the MaskGIT Head:
  - Concern: Questions were raised about the motivation for using the MaskGIT head, its impact on hallucinations in world models, and whether improvements were due to increased parameters.
  - Response: We clarified that hallucinations in autoregressive models can lead to unrealistic predictions, and the MaskGIT head mitigates this by improving sequence modeling capabilities. We provided evidence that the MaskGIT head actually has fewer parameters than the MLP head, indicating that performance gains are due to better inductive biases rather than parameter counts. We also detailed and explored the differences between latent variables generated by the MaskGIT and MLP heads, providing graphs of the two distributions and explicitly showing that the proposed MaskGIT is more confident when predicting state transitions.

- Performance Evaluation and Statistical Analysis:
  - Concern: Some reviewers felt that GIT-STORM's improvements over STORM were marginal and questioned inconsistencies in reported results.
  - Response: We explained that individual environment results can vary due to stochasticity inherent in reinforcement learning tasks. To provide a reliable assessment, we emphasized statistical metrics like mean, median, interquartile mean (IQM), and probability of improvement, following established guidelines in the field. We clarified any discrepancies and updated figures for consistency.

- Clarity and Presentation Enhancements:
  - Concern: Suggestions were made to improve model descriptions, clarify figures, and make the background section more concise.
  - Response: We updated Figure 1 to reflect the model architecture accurately and added detailed explanations of our design choices. We revised Section 2.1 to focus more directly on content relevant to our contributions, enhancing overall clarity.

---

Per-Reviewer Outcomes:

- Reviewer **Lr34**:
  - Outcome: Although we fulfilled all questions and weaknesses, the reviewer did not reply and actively engage in the review.

- Reviewer **JuLK**:
  - Outcome: Raised their score from **3** to **5**. The reviewer expressed openness to further discussion during the next review stage.

- Reviewer **aqah**:
  - Outcome: Raised score from **5** to **6**

- Reviewer **zpZw**:
  - Outcome: Raised score from **5** to **6**

---

We believe that our revisions have effectively addressed the reviewers' concerns and have strengthened the paper's clarity and contribution to the field. Our work advances the integration of continuous actions with discrete latent states in transformer-based world models, providing valuable insights for future research. Such contribution is extremely relevant because almost every existent foundational model is based on a transformer architecture and trained as a sequence modeling architecture. Therefore, in this paper, we try to show that this approach goes beyond the RL community, beyond the video prediction community, for which we report several results. In this paper, we show that Masked Generative Prior improves the sequence modeling capabilities of autoregressive transformers, which touches almost every sub-field of Generative AI.

We kindly ask the area chairs to consider our responses and the improvements made.

Thank you for your consideration!

---

### Meta-Review · Area_Chair_rsyu · 2024-12-22

**Metareview:**

While GIT-STORM shows some promising results in improving world model sequence modeling through masked generative priors, several critical concerns remain insufficiently addressed. The authors' claim of being the first to apply transformer-based world models to continuous actions was overstated, given the existence of TransDreamer. Additionally, there is concerning ambiguity in the MaskGIT head design and sampling process, particularly regarding the gap between training and inference time behavior. The performance gains over STORM appear modest and heavily depend on specific environments, while being still outperformed by DreamerV3 on DMC tasks. Though the authors provided detailed responses, their explanations remain unconvincing, suggesting the work needs further development before publication. I recommend rejecting this paper.

**Additional Comments On Reviewer Discussion:**

During the rebuttal period, reviewers raised concerns, primarily focusing on overclaimed novelty regarding transformer-based models for continuous actions, unclear state mixer design, questionable performance improvements, and ambiguity in the MaskGIT head's mechanism. The authors responded with new ablation studies, clarified novelty claims, and provided technical explanations. While three reviewers were satisfied and raised their scores, it was still not enough to reach the level of acceptance.

---

### Decision · Program_Chairs · 2025-01-22

Reject